# QSPEC: SPECULATIVE DECODING WITH COMPLEMENTARY QUANTIZATION SCHEMES

## ABSTRACT

Quantization has been substantially adopted to accelerate inference and reduce memory consumption of large language models (LLMs). While activation-weight joint quantization speeds up the inference process through low-precision kernels, we demonstrate that it suffers severe performance degradation on multi-step reasoning tasks, rendering it ineffective. We propose a novel quantization paradigm called QSPEC, which seamlessly integrates two complementary quantization schemes for speculative decoding. Leveraging nearly cost-free execution switching, QSPEC drafts tokens with low-precision, fast activation-weight quantization, and verifies them with high-precision weight-only quantization, effectively combines the strengths of both quantization schemes. Compared to high-precision quantization methods, QSPEC empirically boosts token generation throughput by up to $1.80\times$ without any quality compromise, distinguishing it from other low-precision quantization approaches. This enhancement is also consistent across various serving tasks, model sizes, quantization methods, and batch sizes. Unlike existing speculative decoding techniques, our approach reuses weights and the KV cache, avoiding additional memory overhead. Furthermore, QSPEC offers a plug-and-play advantage without requiring any training. We believe that QSPEC demonstrates unique strengths for future deployment of high-fidelity quantization schemes, particularly in memory-constrained scenarios (*e.g.*, edge devices).

## 1 INTRODUCTION

Large language models (LLMs) have demonstrated remarkable abilities across various domains, including mathematics, coding, and planning (Shao et al., 2024b; Guo et al., 2024a; Huang et al., 2024). Nonetheless, their immense scales pose substantial challenges for deployment due to high memory and computational demands, especially in resource-limited scenarios (*e.g.*, inference on edge devices). Quantization has been an effective compression technique to facilitate LLM inference with limited resources (Lin et al., 2024a; Ashkboos et al., 2024; Zhao et al., 2024b; Lin et al., 2024b). By converting high-precision values (*e.g.*, FP16) into their lower-precision counterparts (*e.g.*, INT4), quantization effectively lowers memory and computational requirements, allowing for larger serving batches and model sizes. Furthermore, the reduced memory footprint boosts token generation throughput by accelerating the typically memory-bound autoregressive decoding process (Zhao et al., 2024a).

Based on the quantized objects, recent quantization algorithms can be broadly classified into two categories: weight-only and WXAX: (1) Weight-only quantization, represented by W4A16 (Lin et al., 2024a), quantizes model weights to low precision (e.g., 4-bit) for storage, and then dequantizes them to a higher precision (*i.e.*, FP16) during inference; (2) WXAX methods, such as W4A4 (Ashkboos et al., 2024; Zhao et al., 2024b) and W8A8 (Xiao et al., 2023), simultaneously quantize both weights and activations, and leverage low-precision hardware support for faster execution without dequantizing them to higher precision. Nevertheless, WXAX schemes generally suffer model performance degradation due to more low-precision activations used (as verified in Sec. 2). This poses a tough trade-off between efficacy and efficiency, raising the question:

*"Is there a quantization solution that boosts efficiency while avoiding performance degradation?"*.

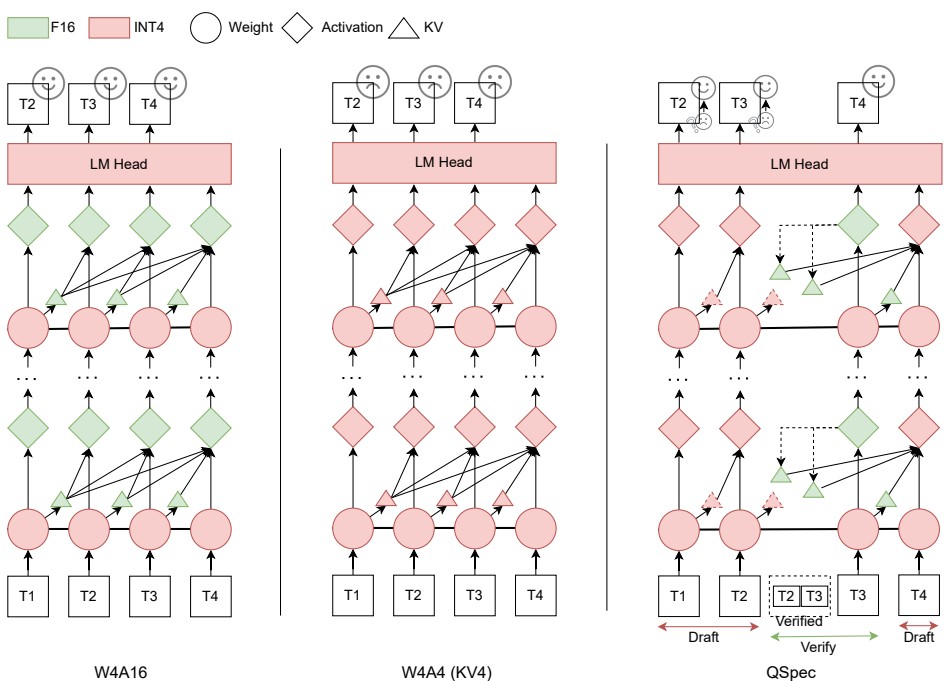

Figure 1: Diagrams of different 4-bit quantization schemes. **Left:** W4A16 uses 4-bit weight and 16-bit activation for inference. **Middle:** W4A4 further adopts 4-bit activation to utilize low-precision W4A4 kernels. **Right:** QSPEC accelerates W4A16 by drafting tokens with W4A4 and verifying them with W4A16, and applies KV cache overwriting for consistent memory consumption.

Considering the comparable performance claims on recent W4A4 methods (Zhao et al., 2024b; Ashkboos et al., 2024), we first contend that their conclusions are biased due to limited evaluation tasks, and W4A4 still experiences significant performance drops when compared to their higher-precision activation counterparts. Specifically, while W4A4 schemes such as Atom (Zhao et al., 2024b) and QuaRot (Ashkboos et al., 2024) perform well on general tasks, such as PIQA (Bisk et al., 2020), Winogrande (Sakaguchi et al., 2019) and ARC Clark et al. (2018), they demonstrate notable performance declines in multi-step reasoning, particularly on mathematical and coding benchmarks (Xiong et al., 2024; Guo et al., 2024b) (shown in Table 1). This raises concerns about the comprehensiveness of evaluation and emphasizes the necessity of incorporating multi-step reasoning tasks into quantization assessment.

Then to answer the above question, we draw inspiration from Speculative Decoding (Leviathan et al., 2023; Chen et al., 2023), which combines rapid drafting of a small model with high-quality token generation of a larger model to boost throughput (*i.e.*, efficiency) without compromising performance (*i.e.*, efficacy). We propose a novel paradigm called QSPEC, which combines mixed-precision quantization execution to **tackle the trade-off between efficiency and efficacy while maintaining the memory usage of high-precision quantization**. Our key insight is that a single weight-quantized model can efficiently toggle two parallel quantization schemes: one with quantized activations and the other without, which we further empirically verify to produce highly similar tokens (Sec. 2.2). This observation unveils the potential for a synergistic approach combining both schemes. As illustrated in Figure 1, for a 4-bit weight-quantized model, we can leverage the faster yet lower-quality execution flow (*i.e.*, W4A4) to draft tokens, while verifying these drafted tokens with the higher-quality quantization flow (*i.e.*, W4A16) with negligible switching costs. Similar to speculative decoding, this 'draft-verify' mode with mixed quantization execution ensures high fidelity with the verifying flow. Differently, our approach re-utilizes the weights and high-precision KV cache, maintaining the memory overhead equivalent to that of the high-precision scheme alone, rather than the sum of both schemes in speculative decoding.

We evaluate the generation quality and end-to-end serving throughput of QSPEC against W4A4 and W4A16 schemes across multiple datasets, model sizes, quantization methods, and batch sizes. Empirically, QSPEC preserves memory consumption and generation quality compared to W4A16,

while offering a high acceptance rate and up to $1.80\times$ higher token generation throughput, thereby mitigating the efficiency-efficacy trade-off of existing quantization methods. Notably, for multi-step reasoning tasks such as MATH (Hendrycks et al., 2021), QSPEC fully compensates for an up to 51.11% decline in generation quality observed with existing W4A4 methods. Furthermore, QSPEC provides plug-and-play compatibility without any training requirements, and can be seamlessly applied to any existing models, delivering superior performance with minimal effort.

In summary, our main contributions are as follows:

- We demonstrate that multi-step reasoning tasks can better capture the performance variations of quantization schemes than current evaluation protocols, and advocate for their incorporation for more comprehensive assessment.

- We validate and instantiate the feasibility of switching between two quantization schemes of a shared weight-quantized model, as well as their high token-level similarities, illuminating future development of quantization schemes.

- We propose QSPEC, synergizing two complementary weight-shared quantization schemes with speculative decoding, alleviating the efficiency-efficacy trade-off of quantization.

- Our empirical results reveal up to $1.80\times$ acceleration without any quality sacrifice across diverse settings. Alongside consistent memory usage, these advantages promise QSPEC for high-fidelity quantization deployment, especially in memory-constrained scenarios.

## 2 MOTIVATION

### 2.1 COMPROMISED PERFORMANCE OF ACTIVATION QUANTIZATION

State-of-the-art (SOTA) activation-weight joint quantization methods, like Atom (Zhao et al., 2024b) and QuaRot (Ashkboos et al., 2024), achieve notable speed-ups with negligible performance loss compared to weight-only ones. However, we argue that this conclusion is skewed by the limited evaluation benchmarks, which fail to capture the negative impacts of activation quantization.

To substantiate this claim, we conduct experiments on Llama-3-8B-Instruct models (Dubey et al., 2024) quantized with W16A16, W4A16, and W4A4 methods across four task datasets: PIQA (Bisk et al., 2020), WikiText-2 (Merity et al., 2016), GSM8K (Cobbe et al., 2021), and MBPP (Austin et al., 2021). PIQA is a two-choice commonsense reasoning benchmark for physical knowledge, evaluated using classification accuracy, while WikiText-2 comprises a collection of high-quality Wikipedia articles, assessed for language fluency via perplexity (Jelinek et al., 1977). Both are commonly adopted in current quantization evaluations. GSM8K includes diverse grade school mathematical problems, evaluated by "exact match" metrics; MBPP focuses on crowd-sourced Python programming challenges, assessed by accuracy. Unlike the former two benchmarks, both GSM8K and MBPP necessitate auto-regressive multi-step reasoning abilities. While these critical abilities are propelled by the rapid advancement in LLMs recently, they have not yet been widely integrated into mainstream quantization evaluation. As shown in Table 1, Atom-based quantization schemes show comparable performance to W16A16 across commonly adopted tasks such as on PIQA and WikiText-2, aligning with the claims in Zhao et al. (2024b). However, W4A4 suffers a nearly 30%

Table 1: Performance of Atom-based quantization schemes with different weight and activation precision across diverse tasks. "Acc", "PPL" and "EM" stand for accuracy, perplexity, and exact match, respectively, with arrows indicating their positive trends. "W16A16" refers to standard FP16 inference, where both weights and activations are represented in FP16 precision.

| Task | Metric | W16A16 | Quantization | |
|---|---|---|---|---|
| | | | Atom (W4A16) | Atom (W4A4) |
| WikiText-2 | PPL $\downarrow$ | 7.73 | 7.87 (+0.15%) | 8.58 (+0.85%) |
| PIQA (10-shot) | EM $\uparrow$ | 78.6 | 77.5 (-1.40%) | 75.6 (-3.81%) |
| MBPP (0-shot) | EM $\uparrow$ | 42.0 | 41.5 (-1.19%) | 30.5 (-27.38%) |
| GSM8K (8-shot) | EM $\uparrow$ | 79.0 | 73.4 (-7.09%) | 54.2 (-31.39%) |

average performance decline on complex reasoning tasks (*i.e.*, on MBPP and GSM8K), whereas W4A16 only experiences about 4%. This indicates that activation quantization leads to several times more performance degradation on multi-step reasoning tasks, despite the improved efficiency. Besides, the performance trend observed on multi-step reasoning tasks shows a stronger correlation with quantization precision than perplexity does, validating their adequacy in assessing quantization performance.

In summary, activation quantization still incurs significant performance loss on more advanced multi-step reasoning tasks. This necessitates the inclusion of reasoning tasks in quantization evaluation for a more comprehensive assessment. On the other hand, this also underscores the demand for a quality-preserving yet efficient quantization paradigm.

## 2.2 HIGH-SIMILARITY TOKEN PREDICTIONS

Despite the notable performance decline caused by activation quantization, we observe, more microscopically, high similarity in top-1 token predictions between quantization schemes with high and low precision activations. Specifically, we first employ Atom-based W4A16 greedy sampling to generate the golden token sequences for the GSM8K test set, obtaining the prediction probabilities for each top-1 answer token. Subsequently, we perform one Atom-based W4A4 forward pass (*i.e.*, prefill) on the concatenated input of each question and its corresponding golden answer to acquire the token probabilities as well. This allows us to assess the prediction discrepancy between W4A4 and W4A16. As illustrated in Figure 2, we observe that (1) the majority of token prediction probabilities of both W4A4 and W4A16 exceed 80%, and most of the tokens associated with high probabilities are accepted. (2) Compared to accepted tokens, the number of rejected ones is negligible, underscoring the high similarity between the two quantization methods. Combined with the analysis in Sec. 2.1, this can be interpreted that a small set of salient token variations can trigger a snowball effect of errors, especially on multi-step reasoning tasks where the subsequent steps are closely conditioned on the previous ones, akin to findings in Zhang et al. (2023), thus impairing the performance of the low-precision activation scheme. Prior studies indicate that low similarity leads to frequent token rejections, thereby diminishing the efficiency of speculative decoding (Leviathan et al., 2023). The observed high token-level similarity suggests that we could potentially restore the generation quality by detecting and correcting a limited number of generation errors incurred by activation quantization. This insight motivates us to propose a quantization-specific speculative decoding framework.

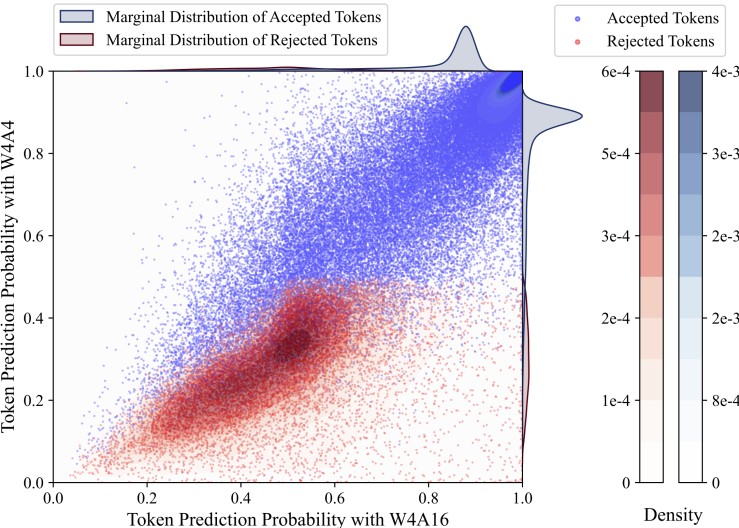

Figure 2: Scatter plot of token prediction probabilities for Atom-based W4A4 and W4A16 on GSM8K test set, along with their two-dimensional and marginal probability distributions. A striking similarity between the two quantization schemes is observed, laying the foundation of QSPEC.

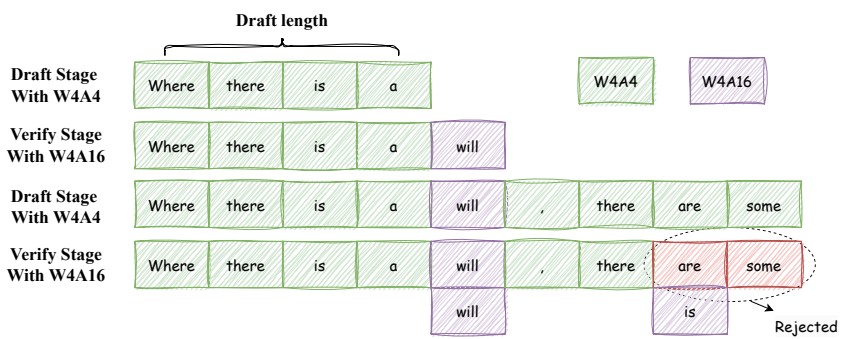

Figure 3: A mini-sample of QSPEC, where green, red, and purple tokens represent draft tokens, rejected tokens, and tokens generated directly by W4A16, respectively.

## 3 METHOD

Targeting an efficient quantization scheme without sacrificing performance or increasing memory consumption, we propose a new quantization paradigm called speculative decoding with complementary quantization execution (QSPEC). As shown in Figure 1, QSPEC employs a draft-verify pipeline for next-token prediction with varying activation precisions and shared low-precision quantized weights, instead of a single quantization scheme. The details are elaborated below, adhering to the core component breakdown of regular speculative decoding (Leviathan et al., 2023).

### 3.1 QSPEC

**Draft Phase.** Current LLMs typically utilize an autoregressive process for next-token prediction, where a new token is drawn from a probability distribution conditioned on all previously generated tokens. This process can be formulated as:

$$t_{i+1} \sim p_{i+1}(t) \coloneqq \mathcal{M}(t_{i+1}|T_{\leq i}), \tag{1}$$

where $\mathcal{M}$ denotes the model including the weight and activation configurations, while $t_{i+1}$ and $T_{\leq i}$ represent the next predicted token and the preceding token sequence $(t_0, t_1, \ldots, t_i)$, respectively.

Compared with previous research (Leviathan et al., 2023; Chen et al., 2023), on one hand, we employ a weight-shared quantization scheme with low-precision activations, rather than one standalone small-sized model, to speculate the next $\gamma$ tokens $\hat{T}_{i+1:i+\gamma}$ and their associated distributions $\hat{p}_{i+1:i+\gamma}(t)$. In $\hat{T}_{i+1:i+\gamma}$, each token $\hat{t}_j$ is sampled from $\mathcal{M}_l(\hat{t}_j|T_{\leq i}, \hat{T}_{i+1:j-1})$, where $j \in [i+1, i+\gamma]$ and $\mathcal{M}_l$ represent our quantized model executed with low-precision activation. On the other hand, our low-precision quantization scheme shares similar attributes as the draft model in Leviathan et al. (2023), as both can generate tokens rapidly, though with reduced quality.

**Verify Phase.** To compensate for the performance decline incurred by excessive quantization, we employ a high-precision weight-only quantization scheme to verify the proposed draft token sequence. This ensures that the final generation quality aligns with that of a high-precision activation quantization scheme. All drafted tokens are verified in parallel for higher efficiency.

Formally, the high-precision quantization scheme $\mathcal{M}_h$ receives as input the concatenation of $T_{\leq i}$ and $\hat{T}_{i+1:i+\gamma}$, producing high-quality prediction probabilities $p_{i+1:i+\gamma+1}(t)$ through a single forward pass. Following this, an acceptance policy $\mathcal{A}$, which will be detailed later, is applied to rectify each drafted token sequentially. Once a token $\hat{t}_{i+j}$ is rejected, all subsequent tokens are discarded, and token $t_{i+j}$ is resampled according to the distribution $p_{i+j}(t)$. In the optimal scenario, all drafted tokens from the low-precision quantized model are accepted by the high-precision model. Subsequently, an additional token $t_{i+\gamma+1}$ is sampled from $p_{i+\gamma+1}(t)$. From this point, a new draft-verify cycle commences, persisting until the sequence is finalized.

**Acceptance Policy.** To maintain high reproducibility, both low-precision and high-precision activation quantization schemes utilize greedy decoding throughout the generation process. This means

that one drafted tokens $\hat{t}_{i+j}$ is accepted as $t_{i+j}$ only when the top-1 tokens from $p_{i+j}$ and $\hat{p}_{i+j}$ coincide; otherwise, this token is rejected. Nonetheless, we claim that alternative strategies, as outlined in Leviathan et al. (2023), can be directly applied to our method due to the similarities in the framework. Figure 3 illustrates a mini-sample of this cycle with the draft token length $\gamma = 4$. The model initially speculates four tokens using W4A4 scheme. Subsequently, adhering to a predefined acceptance policy, it accepts all drafted tokens after verifying them through the W4A16 scheme. In the second loop, however, only the first two tokens are accepted. A new token "is" is directly derived from the prediction probability of W4A16 scheme, and another draft-verify cycle will commence from the ninth token.

**KV Cache Overwriting.** To further reduce memory consumption, QSPEC overwrites KV caches of low-precision activation quantization with those of high-precision method. Specifically, compared to W4A4, W4A16 is expected to yield a higher quality FP16 KV cache due to higher activation precision. Alongside the shared weights, this naturally allows overwriting the low-quality KV caches generated by W4A4 with those from W4A16 for accepted tokens after each validation phase. This enables W4A4 to condition on high-quality KV caches for subsequent autoregressive generation, and saves the memory occupation of W4A4 KV caches, despite requiring negligible buffer space for temporarily storing. To some extent, this operation aligns with the settings of attention kernels in prior works (Zhao et al., 2024b; Shao et al., 2024a; Ashkboos et al., 2024), where INT4 KV caches are typically dequantized to FP16 before or during precision-sensitive attention operations to ensure accurate computations.

## 3.2 ADVANTAGE ANALYSIS

As shown in Table 2, we compare QSPEC with individual quantization schemes (*i.e.*, W4A4 and W4A16) as well as speculative decoding across the dimensions of memory, computation, and generation. QSPEC offers several key advantages over these methods, detailed as follows:

- **Memory-efficient.** Quantization is often motivated by memory constraints, rendering regular speculative decoding unsuitable due to the additional memory allocation for the weights and KV caches of the draft model. However, QSPEC addresses these memory overheads by sharing weights and overwriting KV caches, aligning with the costs associated with standalone high-precision activation quantization.

- **No efficiency-efficacy trade-off.** Leveraging the speculative decoding framework, QSPEC achieves efficiency gains without any quality sacrifice, thereby avoiding the trade-off between efficiency and efficacy. In contrast, individual quantization methods either endure significant performance degradation or accept reduced inference speed.

- **High acceptance rate.** The shared weights inherently enable a strong similarity between the two quantization methods. Besides, the KV cache overwriting further enhances the consistency of subsequent predictions. Both factors collectively contribute to a high token acceptance rate of QSPEC.

- **Plug-and-play compatibility.** Compared to individual quantization schemes, QSPEC simply integrates an acceptance policy and a KV cache overwriting operation. This allows QSPEC to be swiftly implemented based on existing quantization codes without extensive modifications. Furthermore, QSPEC operates without additional training or classifiers, enabling its direct application to any existing model for enhanced inference efficiency.

Table 2: Comparison of individual quantization schemes, regular speculative decoding, and QSPEC across memory, computation, and generation aspects.

| Method | Memory | | Computation | | Generation | |
|---|---|---|---|---|---|---|
| | Draft Weight | Draft KV | W4A4 Kernel | Draft-Verify | High Acceptance Rate | High Fidelity |
| W4A16 | ✗ | ✗ | ✗ | ✗ | - | ✓ |
| W4A4 | ✗ | ✗ | ✓ | ✗ | - | ✗ |
| Speculative Decoding | ✓ | ✓ | ? | ✓ | ? | ✓ |
| QSPEC | ✗ | ✗ | ✓ | ✓ | ✓ | ✓ |

Table 3: Performance of different quantization methods across multiple general and reasoning benchmarks: PIQA, WinoGrande, GSM8K, MATH, MBPP, and HumanEval. The quality degradation ratio is calculated by $\frac{W4A4}{W4A16} - 1$.

| Method | Quantization | WikiText-2[5] PPL ↓ | PIQA EM (%) ↑ | WinoGrande EM (%) ↑ | GSM8K EM (%) ↑ | MATH EM (%) ↑ | MBPP Pass@1 (%) ↑ | HumanEval Pass@1 (%) ↑ |
|---|---|---|---|---|---|---|---|---|
| Atom | W16A16 | 7.73 | 76.8 | 61.4 | 76.2 | 24.9 | 42.5 | 53.0 |
| | W4A16 | 7.87 | 74.8 | 62.0 | 73.4 | 24.3 | 42.0 | 52.4 |
| | QSPEC | 7.87 | 75.0 | 62.0 | 73.4 | 24.3 | 40.5 | 52.4 |
| | W4A4 | 8.6 (+9.58%) | 65.8 (-12.03%) | 56.2 (-9.35%) | 54.7 (-25.47%) | 15.5 (-36.21%) | 33.0 (-21.43%) | 31.7 (-39.50%) |
| QuaRot | W16A16 | 7.73 | 76.8 | 61.4 | 76.2 | 24.9 | 42.5 | 53.0 |
| | W4A16 | 8.58 | 74.2 | 59.4 | 70.5 | 24.7 | 40.0 | 45.7 |
| | QSPEC | 8.58 | 74.4 | 59.2 | 71.0 | 24.7 | 40.5 | 47.6 |
| | W4A4 | 10.2 (+19.24%) | 62.6 (-15.63%) | 53.8 (-9.43%) | 42.0 (-40.43%) | 12.3 (-51.11%) | 28.5 (-28.75%) | 28.0 (-38.73%) |

## 4 EXPERIMENTS

Our evaluation answers three key questions:

Q1: Does QSPEC preserve the quality of high-precision weight-only quantization? (Sec. 4.2)

Q2: Does QSPEC accelerate high-precision weight-only quantization methods? (Sec. 4.3)

Q3: What is the acceptance rate of QSPEC, and the impact of draft token length on it? (Sec. 4.3)

### 4.1 GENERAL SETUP

**Benchmarks.** We assess QSPEC with two primary criteria: (1) generation fidelity and (2) end-to-end serving speedup. For fidelity evaluation, we adopt not only traditional tasks, including PIQA (500, 10-shot) (Bisk et al., 2020), WinoGrande (500, 5-shot) (Sakaguchi et al., 2019), and Wiki-Text2 (Merity et al., 2016), but also challenging multi-step reasoning tasks such as GSM8K (All, 8-shot) (Cobbe et al., 2021), MATH (All, 4-shot) (Hendrycks et al., 2021), MBPP (200, 0-shot) (Austin et al., 2021), and HumanEval (All, 0-shot) (Chen et al., 2021). To measure the acceleration, we use all the above reasoning tasks and two additional chatbot datasets, namely ShareGPT (RyokoAI, 2021) and LMsys-1K (Zheng et al., 2023). Following the setup of Atom (Zhao et al., 2024b), we randomly sampled the dataset for the request prompts to reduce the workload. Due to memory limitations, we vary the batch size from 8 to 32 and serve all requests in a first-come, first-served (FCFS) manner. Once any request is finished, we refill the batch, adhering to the continuous batching approach of ORCA (Yu et al., 2022). We use greedy sampling for token generation.

**Base Models.** To assess the effectiveness and scalability of our approach, we conduct experiments using multiple models from the Llama family (Dubey et al., 2024)[1] with varying scales and capacities: Llama3.2-3b, Llama2-7b, Llama3-8b-instruct, and Llama2-13b.

**Implementation.** All experiments are performed on a node equipped with four NVIDIA A100 GPUs (40GB HBM each) running CUDA 12.5. For the results on NVIDIA L20 GPUs, please refer to Appendix A. To demonstrate the versatility of QSPEC, we implement two SOTA 4-bit quantization methods, namely Atom (Zhao et al., 2024b) and QuaRot (Ashkboos et al., 2024). For W4A16 configurations, we incorporate AWQ-style (Lin et al., 2024a) weight dequantization logic for runtime inference. We select Atom to showcase the acceleration of QSPEC. We use these Group-wise quantization schemes with a group size of 128. With the draft token length $\gamma$ as 3, we simulate the performance of QSPEC by initially employing fake quantization to fully emulate the execution flow, encompassing both the draft and verify stages of QSPEC. Subsequently, we replay the collected traces with real kernel execution to accurately reproduce the latency.[2]

### 4.2 FIDELITY EVALUATION

**QSPEC effectively maintains the generation quality of W4A16, whereas W4A4 does not.** As listed in Table 3, with the draft verification of W4A16, QSPEC exhibits only minimal performance

---

[1] https://www.huggingface.co/meta-llama

[2] Atom's kernel only supports shape-specific models. We modify the model structure to meet requirements while maintaining the original model size.

Table 4: Comparison of token generation throughput across different model sizes, quantization configurations, and batch sizes for various datasets. All values are measured in token/s. "Avg." denotes the average speedup ratio for the corresponding row or column.

| Model | Method | Batch | GSM8K | MATH | MBPP | HumanEval | ShareGPT | LMsys-1k | Avg. |
|---|---|---|---|---|---|---|---|---|---|
| 3B[1] | W4A16 | 8 | 326.1 | 360.2 | 429.0 | 392.4 | 434.7 | 401.7 | – |
| | | 16 | 529.1 | 653.8 | 820.3 | 673.2 | 882.2 | 772.9 | – |
| | | 32 | 679.2 | 906.8 | 1261.4 | 848.4 | 1439.3 | 1103.6 | – |
| | QSPEC | 8 | 501.5 (1.54×) | 537.2 (1.49×) | 640.4 (1.49×) | 593.1 (1.51×) | 646.6. (1.49×) | 592.7 (1.48×) | 1.50× |
| | | 16 | 731.9 (1.38×) | 837.8 (1.28×) | 863.2 (1.17×) | 875.9 (1.30×) | 1081.4 (1.23×) | 945.0 (1.22×) | 1.24× |
| | | 32 | 950.3 (1.40×) | 1175.8 (1.30×) | 1371.0 (1.09×) | 1052.3 (1.24×) | 1645.9 (1.14×) | 1347.6 (1.22×) | 1.23× |
| | Avg. | | 1.44× | 1.36× | 1.25× | 1.35× | 1.29× | 1.31× | 1.33× |
| 7B | W4A16 | 8 | 126.8 | 144.1 | 165.0 | 170.4 | 177.1 | 157.0 | – |
| | | 16 | 213.1 | 267.2 | 314.9 | 344.6 | 358.6 | 300.8 | – |
| | | 32 | 257.6 | 347.1 | 409.7 | 478.9 | 509.1 | 402.0 | – |
| | QSPEC | 8 | 203.7 (1.61×) | 239.9 (1.66×) | 234.8 (1.42×) | 281.5 (1.65×) | 274.4 (1.55×) | 241.4 (1.54×) | 1.57× |
| | | 16 | 312.3 (1.47×) | 377.6 (1.41×) | 380.1 (1.21×) | 459.6 (1.33×) | 455.2 (1.27×) | 379.5 (1.26×) | 1.33× |
| | | 32 | 496.2 (1.54×) | 488.5 (1.41×) | 473.4 (1.16×) | 620.2 (1.30×) | 633.1 (1.24×) | 498.3 (1.24×) | 1.31× |
| | Avg. | | 1.54× | 1.50× | 1.26× | 1.43× | 1.35× | 1.35× | 1.40× |
| 8B | W4A16 | 8 | 121.8 | 131.2 | 155.2 | 153.4 | 163.8 | 152.4 | – |
| | | 16 | 210.3 | 247.0 | 300.7 | 293.5 | 365.6 | 311.2 | – |
| | | 32 | 277.1 | 355.3 | 425.1 | 398.7 | 619.1 | 486.5 | – |
| | QSPEC | 8 | 191.7 (1.57×) | 200.4 (1.53×) | 214.4 (1.38×) | 230.1 (1.50×) | 241.0 (1.47×) | 220.6 (1.45×) | 1.48× |
| | | 16 | 294.2 (1.40×) | 333.4 (1.35×) | 334.1 (1.11×) | 373.0 (1.27×) | 431.7 (1.18×) | 373.4 (1.20×) | 1.25× |
| | | 32 | 368.8 (1.33×) | 447.5 (1.26×) | 478.1 (1.12×) | 484.2 (1.21×) | 687.3 (1.11×) | 564.1 (1.16×) | 1.20× |
| | Avg. | | 1.43× | 1.38× | 1.21× | 1.33× | 1.25× | 1.27× | 1.31× |
| 13B[1] | W4A16 | 8 | 74.0 | 85.1 | 103.7 | 100.5 | 104.1 | 92.3 | – |
| | | 16 | 128.6 | 163.0 | 185.8 | 177.7 | 222.8 | 173.1 | – |
| | | 32 | 195.1 | 206.9 | 323.7 | 327.8 | 330.1 | 241.6 | – |
| | QSPEC | 8 | 127.8 (1.73×) | 148.6 (1.75×) | 173.4 (1.67×) | 180.8 (1.80×) | 173.5 (1.68×) | 150.2 (1.63×) | 1.71× |
| | | 16 | 194.7 (1.51×) | 235.4 (1.44×) | 285.9 (1.54×) | 292.5 (1.65×) | 288.7 (1.30×) | 222.9 (1.29×) | 1.45× |
| | | 32 | 247.1 (1.27×) | 307.4 (1.49×) | 399.9 (1.24×) | 435.3 (1.33×) | 407.1 (1.23×) | 323.3 (1.34×) | 1.31× |
| | Avg. | | 1.50× | 1.56× | 1.48× | 1.59× | 1.40× | 1.42× | 1.49× |

fluctuations compared to W4A16. This negligible variation may stem from the nondeterministic algorithms of PyTorch[3] or occasional cases where two tokens have the same maximum prediction probability. In contrast, W4A4 experiences a substantial performance decline exceeding 10% across most tasks, with the reduction becoming more pronounced as task difficulty increases. For instance, compared to GSM8K and MBPP, the performance drop for W4A4 is much greater on the more challenging MATH and HumanEval tasks, showing declines of 51.11% and 38.73%, respectively. On the other hand, this also highlights the higher sensitivity of multi-step reasoning tasks to the negative effects of quantization compared to regular tasks, such as WikiText-2[4] and WinoGrande. This observation aligns with our earlier analysis in Sec. 2, encouraging incorporating multi-step reasoning tasks into quantization evaluation.

## 4.3 ACCELERATION EVALUATION

**QSPEC exhibits a substantial efficiency boost compared to W4A16.** In Table 4, we present the token generation throughput for both QSPEC and W4A16 across different model sizes, quantization configurations, and batch sizes on diverse datasets. On average, QSPEC achieves a throughput increase of $1.38\times$ over W4A16 across all settings, with a peak improvement of $1.80\times$.

**Speedup does not show an evident correlation with task difficulty.** As shown at the bottom of Table 4, we calculate the average acceleration ratios across all configurations for each dataset. When comparing on simpler dialogue datasets (*i.e.*, ShareGPT and LMsys-1k), QSPEC exhibits negligible throughput variation on multi-step reasoning tasks, particularly on coding tasks, despite a more pronounced performance decline of W4A4. Even on GSM8K and MATH tasks, a higher throughput is observed, due to the application of 8-shot and 4-shot prompts, respectively. This finding supports

---

[3] https://pytorch.org/docs/stable/notes/randomness.html
[4] To measure perplexity, only a single prefill stage is necessary, bypassing the verify stage.

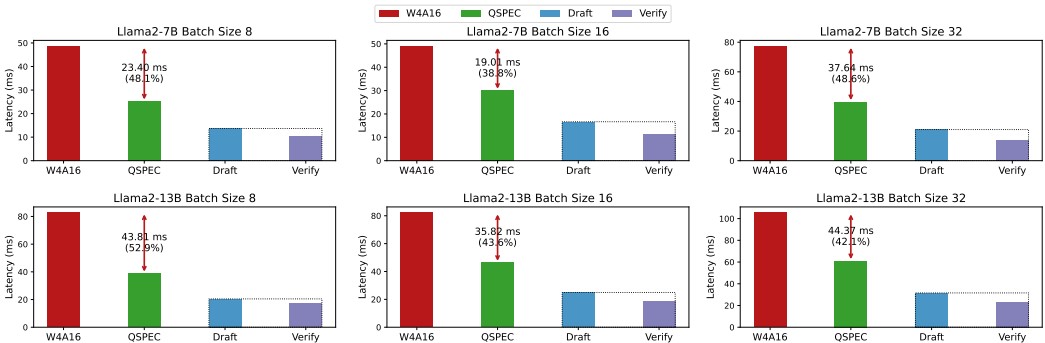

Figure 4: Per-valid-token latency comparison of QSPEC and W4A16 across different models and batch sizes. The latency of QSPEC is further decomposed into draft and verify categories.

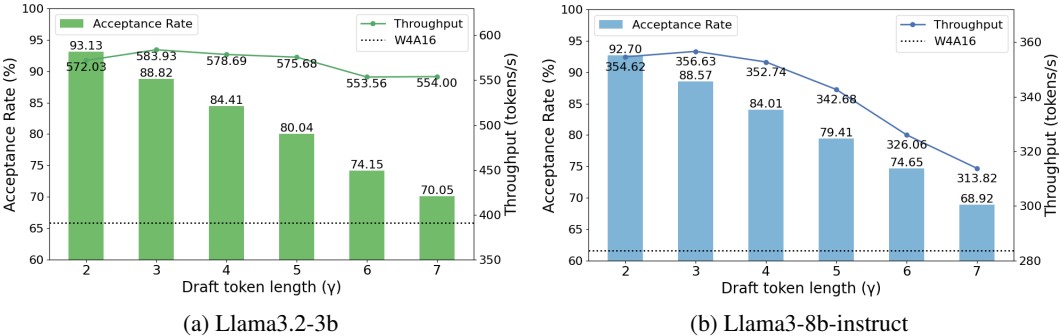

(a) Llama3.2-3b     (b) Llama3-8b-instruct

Figure 5: Acceptance rate and throughput of Llama3.2-3b (batch size 8) and Llama3-8b-instruct (batch size 16) with respect to the draft token length $\gamma$.

our observations in Sec. 2, where we attribute the reduction in multi-step reasoning capabilities to a few token changes that cause a worsening snowball effect, rather than numerous token prediction errors.

**Larger models tend to yield better speedup ratios.** We compare the average acceleration on all datasets across different models, and find a gradually-increasing acceleration as the base model scales up[5]. This overall upward trend indicates a promising outlook for our approach with larger models, although further experiments are needed for confirmation. Due to resource limitations, this will be addressed in future work.

**Latency Composition.** As illustrated in Figure 4, we compute the per-valid-token latency by dividing the total latency by only the number of accepted tokens before averaging on all evaluation datasets. Notably, QSPEC achieves remarkable latency savings ranging from 38.8% to 52.9%. Besides, the per-token latency is further decomposed into two components: draft and verify latency. Clearly, the primary gains of QSPEC arise from the rapid drafting capability and the reduced latency achieved through the parallel verification of multiple tokens.

**Ablation on Draft Token Length.** To assess parameter sensitivity, we vary the draft token lengths $\gamma$, the sole hyper-parameter of QSPEC, from 2 to 7 across all the benchmarks using Llama3.2-3b and Llama3-8b-instruct models. As depicted in Figure 5, an increase in $\gamma$ leads to a gradual decline in the token acceptance rate, since all subsequent tokens are discarded once a token is rejected. Nevertheless, even at $\gamma = 7$, the token acceptance rate remains relatively high, approximately 70%, compared to $28 \sim 58\%$ in 160m-7b draft-target model pair under $\gamma = 5$ in conventional specula-

---

[5]It is noteworthy that Llama2-7B shows higher speedup than Llama3-8B. This stems from the size difference primarily related to vocabulary, coupled with the introduction of Group-Query Attention (Ainslie et al., 2023), reducing the computation workload.

tive decoding (Liu et al., 2024). Additionally, a consistent improvement in throughput is observed compared to W4A16, indicating the robustness of QSPEC with respect to $\gamma$.

## 5 RELATED WORK

**Quantization** is a common technique for deploying LLMs on resource-limited scenarios. Broadly, recent quantization algorithms can be classified into two categories: weight-only W4A16 and weight-activation joint W4A4. Notably, AWQ (W4A16) (Lin et al., 2024a) redistributes the quantization burden by scaling salient weight channels to protect them from degradation. In contrast, W4A4 aggressively quantizes activations to leverage low-precision hardware for improved speed at the cost of model quality degradation. To address this challenge, Atom (Zhao et al., 2024b) proposes reordering outlier channels in the activation through offline profiling. Similarly, QuaRot (Ashkboos et al., 2024) employs Hadamard matrices to apply computational invariance on weights. Despite these advancements, our observations indicate that W4A4 methods still exhibit substantial degradation compared to weight-only quantization approaches across multi-step reasoning tasks. On the other hand, *adaptive quantization* aims to optimize the trade-off between quantization-induced quality degradation and computational acceleration by mixed precision. LLM-PQ (Zhao et al., 2024a) proposes an adaptive layer-wise bitwidth selection approach, while QAQ (Dong et al., 2024) focuses on KV-cache bitwidth optimization. Other works operate at finer granularity to address outliers (Lee et al., 2024). However, these methods cannot fully recover the generation quality of higher precision.

**Speculative Decoding** leverages a draft model to generate candidate tokens, which are then validated by a target model (Leviathan et al., 2023). Recent research has primarily focused on improving the acceptance rate and generation speed of candidate tokens. SpecInfer (Miao et al., 2024) introduces a boost-tuned small language model to generate candidate tokens in tree structures, enabling single-pass verification. In contrast, EAGLE (Li et al., 2024) adopts an aggressive pruning strategy for the draft model's architecture, allowing penultimate layer feature prediction with minimal computational overhead. Self-speculative decoding, a subset of this technique, employs a single model for both draft generation and verification. LayerSkip (Elhoushi et al., 2024) introduces a training methodology for early exit with layer drop, subsequently verifying partially generated tokens through full model inference. Medusa (Cai et al., 2024) constructs a generation tree of multiple candidate continuations by augmenting the original LLM with additional heads atop the final hidden state while relaxing the acceptance policy. However, these approaches inevitably require retraining of the original model, which can be computationally expensive and time-consuming.

**Parameter Sharing** has been extensively applied for various purposes in previous research. Targeting parameter savings, Universal Transformer (Dehghani et al., 2018) shares all layers within a transformer model, while Subformer (Reid et al., 2021) shares its middle layers without sacrificing performance. Similarly, DictFormer (Lou et al., 2021) reparameterizes the model using a shared dictionary alongside unshared coefficients and indices, achieving reduced parameter redundancy and faster computations. Pires et al. (2023) enhances both accuracy and latency by implementing a single, larger shared feed-forward network across the encoder. In a different domain, Wang et al. (2024b;a) and Kopiczko et al. (2023) leverage parameter sharing in low-rank adaptation (LoRA) (Hu et al., 2021) to improve parameter efficiency. Unlike these methods, our focus is on sharing low-precision weights from two quantization schemes to maintain memory overhead.

## 6 CONCLUSION

In this paper, we begin by validating that multi-step reasoning tasks can capture performance degradation incurred by activation quantization more sensitively and consistently than current evaluation protocols, advocating for their incorporation for a more comprehensive assessment. With nearly cost-free execution switching and high token-level similarities, we introduce QSPEC, a novel quantization paradigm that seamlessly synergizes two complementary weight-shared quantization schemes with speculative decoding. Empirically, QSPEC achieves up to $1.80\times$ acceleration without any quality sacrifice across diverse settings. Alongside consistent memory consumption and a plug-and-play property, these advantages distinguish QSPEC from any existing solution, promising it for high-fidelity quantization deployment, particularly in memory-constrained scenarios.

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

# A SUPPLEMENTARY EXPERIMENTS

We further extend our experiments on NVIDIA L20 GPUs, and complement additional analysis of W16A16 (Wolf et al., 2020), Atom-based W4A16 (Lin et al., 2024a), W4A4 (Zhao et al., 2024b), and QSPEC.

**Consistent Efficiency Enhancement of QSPEC over W4A16.**   As presented in Table 5, we detail the token generation throughput for both QSPEC and WXAX methods across various model sizes, quantization configurations, batch sizes, and datasets. Compared to W4A16, QSPEC achieves a throughput increase of $1.33\times$ across all the settings on average, with a peak improvement of $1.64\times$. These results, along with those in Table 4, validate the consistent efficiency superiority of QSPEC over W4A16 on different GPU platforms. Additionally, QSPEC consistently outperforms W16A16 in terms of efficiency across all the settings.

**Preserved Generation Quality of QSPEC Compared to W4A16.**   As illustrated in Figure 6, we visualize the generation quality (i.e., accuracy) and efficiency (i.e., throughput). Aligning with the analysis of Table 1, W4A4 experiences a significant performance decline, ranging from 18.5% to 39.5%, on multi-step reasoning benchmarks when compared to W4A16. In contrast, QSPEC not only maintains the performance of W4A16 (slightly lower than that of W16A16 due to weight quantization for memory saving), but also offers much higher throughput.

**Detailed Latency Decomposition of Per Valid Token.**   As shown in Figure 7, we calculate the per-valid-token latency by dividing the total latency by the number of accepted tokens in each sample, which is then averaged across all samples and evaluation datasets. Notably, the decode stage accounts for the majority of the time latency when compared to the prefill stage. With the rapid drafting capability and parallel verification, QSPEC achieves significantly lower latency than W4A16, ranging from 28.5% to 39.7%. In detail, QSPEC spends more time in the draft phase than in the high-precision verify phase. This may be attributed to the high acceptance rate of QSPEC, which resulted in less verify requests.

**Ablation on Draft Token Length.**   To assess parameter sensitivity, we vary the draft token length $\gamma$, the sole hyperparameter of QSPEC, from 2 to 7 across all benchmarks with Llama3.2-3b and Llama3-8b-instruct models. For a thorough comparison, we also include the throughput of W16A16 and W4A16 as references. As depicted in Figure 8, an increase in $\gamma$ results in a gradual decrease in the token acceptance rate, since the rejection of any token leads to the discarding of all subsequent tokens. Nevertheless, even at $\gamma = 7$, the token acceptance rate remains relatively high at approximately 70%, compared to the 28%–58% observed in the 160m–7b draft-target model pair under $\gamma = 5$ in conventional speculative decoding (Liu et al., 2024). Additionally, we observe a continuous improvement in throughput compared to W4A16, indicating the hyperparameter robustness of QSPEC. With an appropriate choice of $\gamma$ (*i.e.*, $\gamma \leq 5$), QSPEC consistently outperforms W16A16 in both memory consumption and efficiency.

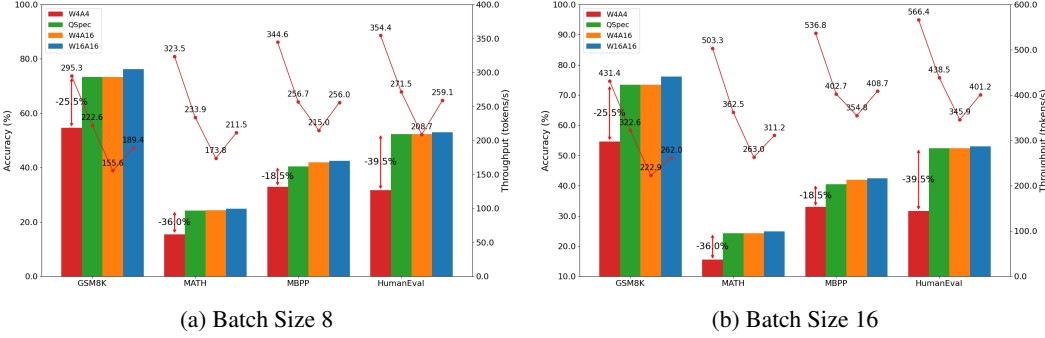

(a) Batch Size 8                                    (b) Batch Size 16

Figure 6: Comparison of accuracy and efficiency among W16A16, W4A16, W4A4, and QSPEC across various datasets with batch sizes of 8 and 16, respectively. The bars and lines represent the accuracy and throughput of each method.

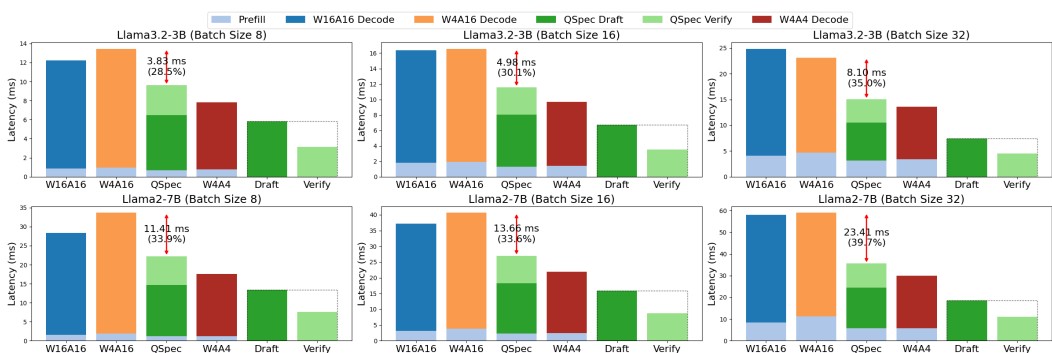

Figure 7: Per-valid-token latency decomposition of W16A16, W4A16, QSPEC and W4A4 across different models and batch sizes. The latency of QSPEC is further decomposed into draft and verify categories for details.

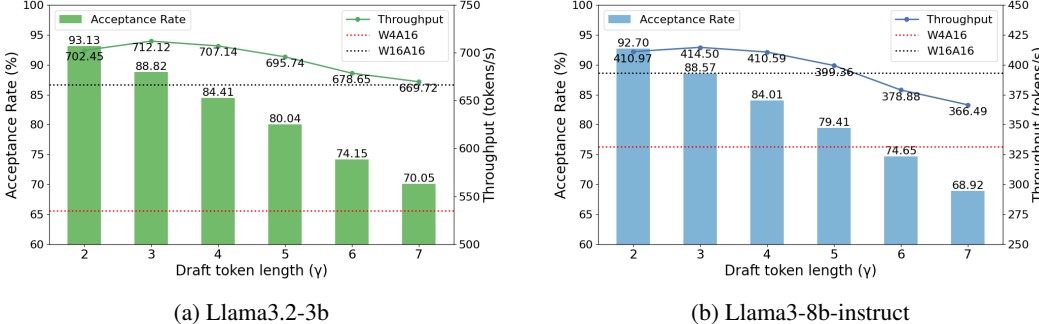

(a) Llama3.2-3b                    (b) Llama3-8b-instruct

Figure 8: Acceptance rate and throughput of Llama 3.2-3b (with a batch size of 8) and Llama 3-8b-instruct (with a batch size of 16) with respect to the draft token length $\gamma$.

Table 5: Comparison of token generation throughput across different model sizes, quantization configurations, and batch sizes for various datasets. All values are measured in token/s. "Avg." denotes the average speedup ratio for the corresponding row or column. "†" indicates the failure of W4A16 kernels to support these batch sizes together with long sequences and the large models.

| Model | Method | Batch | GSM8K | MATH | MBPP | HumanEval | ShareGPT | LMsys-1k | Avg. |
|---|---|---|---|---|---|---|---|---|---|
| 3B[1] | W16A16 | 8 | 511.1 | 588.7 | 756.6 | 647.2 | 785.7 | 711.2 | – |
| | | 16 | 666.5 | 845.6 | 1171.0 | 948.3 | 1292.2 | 1126.4 | – |
| | | 32 | 833.4 | 1081.5 | 1697.7 | 1111.6 | 1975.6 | 1553.3 | – |
| | W4A4 | 8 | 804.7 | 921.2 | 1002.0 | 892.6 | 1091.6 | 990.3 | – |
| | | 16 | 1109.1 | 1374.5 | 1548.0 | 1289.8 | 1763.5 | 1581.0 | – |
| | | 32 | 1424.3 | 1899.3 | 2300.6 | 1488.2 | 2777.3 | 2194.4 | – |
| | W4A16 | 8 | 420.0 | 476.7 | 604.5 | 535.7 | 610.4 | 559.8 | – |
| | | 16 | 578.5 | 715.9 | 989.7 | 804.4 | 1080.2 | 925.8 | – |
| | | 32 | 726.3 | 933.8 | 1536.7 | 954.4 | 1704.5 | 1336.4 | – |
| | QSPEC | 8 | 594.1 (1.41×) | 648.2 (1.36×) | 760.1 (1.26×) | 723.6 (1.35×) | 787.5 (1.29×) | 738.8 (1.32×) | 1.33× |
| | | 16 | 811.5 (1.40×) | 936.0 (1.31×) | 1157.8 (1.17×) | 1042.1 (1.30×) | 1294.5 (1.20×) | 1171.4 (1.27×) | 1.27× |
| | | 32 | 1030.4 (1.42×) | 1240.2 (1.33×) | 1617.4 (1.05×) | 1248.5 (1.31×) | 1969.6 (1.16×) | 1576.0 (1.18×) | 1.24× |
| | Avg. | | 1.41× | 1.33× | 1.16× | 1.32× | 1.21× | 1.25 × | 1.28× |
| 7B | W16A16 | 8 | 213.4 | 254.3 | 278.8 | 316.7 | 322.4 | 285.3 | – |
| | | 16 | 290.3 | 362.1 | 447.7 | 505.1 | 541.3 | 441.6 | – |
| | | 32 | 340.9 | 441.6 | 585.3 | 663.6 | 735.3 | 564.2 | – |
| | W4A4 | 8 | 349.5 | 411.7 | 396.1 | 471.2 | 471.8 | 419.4 | – |
| | | 16 | 496.6 | 612.2 | 614.3 | 749.5 | 760.9 | 642.6 | – |
| | | 32 | 620.0 | 793.6 | 801.5 | 1043.9 | 1083.2 | 865.5 | – |
| | W4A16 | 8 | 165.0 | 193.1 | 224.5 | 240.2 | 243.5 | 220.2 | – |
| | | 16 | 231.8 | 286.5 | 384.4 | 407.3 | 435.9 | 358.0 | – |
| | | 32 | 268.9 | 359.9 | 480.0 | 555.9 | 620.2 | 470.1 | – |
| | QSPEC | 8 | 253.7 (1.54×) | 291.5 (1.51×) | 298.3 (1.33×) | 350.9 (1.46×) | 345.7 (1.42×) | 310.3 (1.41×) | 1.44× |
| | | 16 | 359.8 (1.55×) | 420.2 (1.47×) | 466.7 (1.21×) | 555.2 (1.36×) | 557.8 (1.28×) | 473.1 (1.32×) | 1.37× |
| | | 32 | 441.8 (1.64×) | 527.2 (1.46×) | 575.3 (1.20×) | 749.4 (1.35×) | 770.0 (1.24×) | 628.4 (1.34×) | 1.39× |
| | Avg. | | 1.58× | 1.48× | 1.25× | 1.39× | 1.31× | 1.36× | 1.39× |
| 8B | W16A16 | 8 | 189.4 | 211.5 | 256.0 | 259.1 | 290.7 | 265.8 | – |
| | | 16 | 262.0 | 311.2 | 408.7 | 401.2 | 511.0 | 447.4 | – |
| | | 32 | 303.8 | 390.8 | 566.3 | 522.6 | 820.0 | 649.8 | – |
| | W4A4 | 8 | 295.3 | 323.5 | 344.6 | 354.4 | 395.9 | 366.8 | – |
| | | 16 | 431.4 | 503.3 | 536.8 | 566.4 | 697.5 | 621.1 | – |
| | | 32 | 532.8 | 688.5 | 755.7 | 763.7 | 1167.9 | 956.8 | – |
| | W4A16 | 8 | 155.6 | 173.8 | 215.0 | 208.7 | 231.1 | 215.6 | – |
| | | 16 | 222.9 | 263.0 | 354.8 | 345.9 | 422.8 | 369.4 | – |
| | | 32 | † | † | 509.8 | 468.7 | 706.0 | 580.5 | – |
| | QSPEC | 8 | 222.6 (1.43×) | 233.9 (1.35×) | 256.7 (1.19×) | 271.5 (1.30×) | 285.0 (1.23×) | 268.3 (1.24×) | 1.29× |
| | | 16 | 322.6 (1.45×) | 362.5 (1.38×) | 402.7 (1.14×) | 438.5 (1.27×) | 507.5 (1.20×) | 453.5 (1.23×) | 1.28× |
| | | 32 | 400.2 (†) | 362.5 (†) | 578.1 (1.13×) | 573.0 (1.22×) | 798.8 (1.13×) | 684.5 (1.18×) | 1.27× |
| | Avg. | | 1.44× | 1.36× | 1.15× | 1.26× | 1.19× | 1.22 × | 1.27× |
| 13B[1] | W16A16 | 8 | 121.9 | 146.6 | 183.1 | 182.0 | 187.1 | 160.1 | – |
| | | 16 | 169.6 | 211.2 | 304.4 | 291.0 | 311.0 | 243.0 | – |
| | | 32 | 202.4 | 253.8 | 426.0 | 423.5 | 311.0 | 334.2 | – |
| | W4A4 | 8 | 194.7 | 228.2 | 253.6 | 261.5 | 259.8 | 228.2 | – |
| | | 16 | 288.3 | 349.2 | 415.3 | 424.9 | 431.5 | 348.4 | – |
| | | 32 | 369.8 | 469.9 | 606.7 | 665.4 | 431.5 | 508.8 | – |
| | W4A16 | 8 | 94.8 | 112.9 | 143.4 | 140.0 | 146.7 | 127.9 | – |
| | | 16 | 136.1 | 171.9 | 250.8 | 236.9 | 255.9 | 207.2 | – |
| | | 32 | † | † | 376.4 | 365.5 | 255.9 | 287.4 | – |
| | QSPEC | 8 | 148.2 (1.56×) | 167.9 (1.49×) | 193.6 (1.35×) | 201.2 (1.44×) | 194.5 (1.33×) | 174.0 (1.36×) | 1.42× |
| | | 16 | 212.8 (1.56×) | 248.6 (1.45×) | 316.8 (1.26×) | 323.3 (1.36×) | 327.4 (1.28×) | 266.9 (1.29×) | 1.29× |
| | | 32 | 266.6 (†) | 320.0 (†) | 451.5 (1.20×) | 483.0 (1.32×) | 327.4 (1.28×) | 379.3 (1.32×) | 1.32× |
| | Avg. | | 1.56× | 1.47× | 1.27× | 1.37× | 1.29× | 1.32× | 1.38× |

