# OpenReview forum: "QSpec: Speculative Decoding with Complementary Quantization Schemes"
_ICLR.cc/2025/Conference — Submitted to ICLR 2025_

### Official Review · Reviewer_FprC · 2024-10-27

**Soundness:** 3
**Presentation:** 3
**Contribution:** 2
**Rating:** 6
**Confidence:** 3

**Summary:**

The authors investigate the impact of LLM quantization on multi-step reasoning tasks and find that current methods are insufficient in some cases. They introduce a system design for speculative decoding where activation quantization is disabled during the verification stage. They find that this can improve HW performance without degrading model quality.

**Strengths:**

Improving the HW performance of multi-step reasoning is a relevant and impactful problem. The authors provide a very insightful analysis into both the impact of quantization on multi-step reasoning and the impact of activation quantization on token prediction probabilities, both of which are novel insights to my knowledge.

**Weaknesses:**

While the results are promising, the benchmarking may be incomplete. As an alternative to this approach, one could just use a smaller draft model without weight sharing assuming memory is not a constraint. Often the significantly smaller draft model is tuned to the verification model to improve acceptance rates. There is no comparison against this dual-model architecture to establish a concrete baseline. As such, it is unclear which approach one should take.

**Questions:**

Have you considered the scenario where a draft model is significantly smaller than the verification model and fine-tuned specifically to maximize acceptance rates? Are the memory benefits from weight sharing still large enough to warrant using QSpec over such an approach?

---

> ### Author Response · Authors · 2024-11-23
> **Response to reviewer (1/n)**
>
> Dear Reviewer FprC,
>
> We sincerely appreciate your detailed observations and valuable feedback on our work! We are motivated by your recognition on our analysis into both (a) the impact of quantization on multi-step reasoning and (2) the impact of activation quantization on token prediction probabilities. In response to your confusion, we want to first highlight the potential future impact of QSpec from both academic and industrial perspectives, followed by answering your questions one by one. Hope that these clarifications could further address your concerns!
>
> **Potential impact of QSpec.**
>
> **1. From the academic perspective.**
> - **QSpec validates the feasibility of nearly cost-free switching** between two quantization schemes
> of a shared weight-quantized model, as well as their high token-level similarities. This may directly illuminate
> the future development of quantization schemes.
> - **QSpec decouples the speed enhancement and quality preservation in existing quantization schemes.** Previously, researchers had to devote their efforts to balance performance loss against acceleration. Now, QSpec assigns the roles of speed improvement and performance assurance to the draft model and verify model, respectively. This separation shifts researchers' focus from the challenging balance to a concentrated effort on acceleration. This new perspective unlocks significant opportunities for further research.
>
> **2. From the industrial perspective.**
> - **QSpec prompts chip vendors to reconsider their design.** Due to the compromised performance of low-precision activation quantization, **many modern NPUs (e.g., Ascend and GraphCore) primarily utilize FP16/BF16 computation cores, resulting in the underutilization of low-precision kernels.** However, QSpec offers a general plug-and-play solution that harnesses low-precision architectures to enhance efficiency without sacrificing quality. Notably, we have received feedback from one chip vendor, indicating that QSpec has inspired them to rethink the proposition of investment that they should allocate to low-bit cores and instructions.
>
> BTW, a recent paper titled "Scaling Laws for Precision" demonstrates that performance degradation becomes increasingly severe as the number of quantization bits decreases, particularly in activation quantization [1]. This finding undoubtedly hampers the development of further quantization schemes. However, as mentioned above, QSpec decouples speed enhancement and quality preservation in quantization methods, focusing quantization efforts on acceleration rather than the tough balance against performance loss. This may foster the continuous development of quantization methods.

---

> > ### Author Response · Authors · 2024-11-23
> > **Response to reviewer (2/n)**
> >
> > **Question 1: Comparison with dual-model speculative methods.**
> > - As highlighted in our Introduction section and echoed by Reviewer 9onb, "the authors propose a weight-sharing speculative decoding framework for quantized models with varying activation precisions." Instead of integrating quantization into speculative decoding, **QSpec is more like a new quantization paradigm**, which leverages the spirit of speculative decoding to solve the problems met in current quantization schemes.
> > - As mentioned in the potential impacts, **QSpec introduces indispensable advantages for quantization methods (e.g., efficiency-performance decoupling) and hardware design**, differentiating it from regular dual-model speculative decoding.
> > - Typically, quantization methods are deployed in environments (e.g., edge devices) with severely limited resources, particularly regarding memory. This eliminates almost all existing speculative decoding, due to increased weight and KV cache consumption. In contrast, the design of QSpec, including both weight-sharing and KV cache overwriting, circumvents the extra memory overheads over W4A16, **aligning with the quantization scenarios**. Together with the above analysis, these advantages make comparisons with regular memory-extensive speculative decoding approaches not that necessary, due to their deviation from the initial motivation of our work.
> > - Besides, QSpec enjoys the following advantages, compared to existing speculative decoding methods:
> >   - **QSpec is completely orthogonal to existing speculative decoding methods.** Specifically, QSpec can seamlessly integrate with existing speculative schemes using a hierarchical structure, such as "(Draft model ➔ W4A4) ➔ W4A16" or "(W4A4 ➔ W4A16) ➔ verify model". This integration will provide additional acceleration benefits for both QSpec and existing solutions.
> >   - **QSpec achieves high acceptance rates.** As shown in Figure 5, Figure 8 in our updated submission and the Table below, QSpec inherently achieves high acceptance rates across diverse draft token length, quantization methods and datasets due to its design of shared weights and KV cache, eliminating the need of finetuning. Besides, together with the efficiency-performance decoupling brought by QSpec, this natively high acceptance rate may illuminate the future development of more efficient quantization methods. In contrast, acceptance rates of 28–58% are observed on a 160m-7b draft-target model pair with γ=5 in conventional speculative decoding [2].
> >   - **QSpec is plug-and-play.** It requires no finetuning and extensive modifications to serve existing models, enabling seamless integrations with them.
> >
> > | Quantization | lmsys | sharegpt | gsm8k | MATH  | MBPP  | HumanEval |
> > |--------------|-------|----------|-------|-------|-------|-----------|
> > | Atom         | 0.867 | 0.838    | 0.915 | 0.894 | 0.886 | 0.914     |
> > | QuaRot       | 0.830 | 0.816    | 0.890 | 0.889 | 0.854 | 0.882     |
> >
> >
> > **Question 2: Enough memory benefits from weight sharing.**
> > - As mentioned above, quantization is typically deployed in environments (e.g., edge devices) with severely limited resources, particularly regarding memory. These are exactly the scenarios that QSpec aims to serve. Weight-sharing is a fundamental component of QSpec, providing the following two indispensable advantages:
> >   - **Reduced weight consumption.** Shared weights eliminate the need to store separate weight parameters for the draft and verify models, significantly lowering memory usage.
> >   - **KV cache reduction.** Weight-sharing is a prerequisite for "KV cache overwriting" between the draft and verify models. Specifically, without weight sharing, speculative decoding must maintain separate KV caches, resulting in substantial memory overhead, particularly for long-context and multi-step reasoning tasks.
> >
> >
> > Hence, compared to deploying a small draft model, weight sharing not only saves weight memory, but also significantly reduces KV cache memory consumption, which can never be achieved by two separate models, even if the draft model is small with the sacrifice of acceptance rate.
> >
> > We sincerely hope that our clarifications could address your concerns, provide a clearer understanding of the impact of our work, and help you reconsider the scores! If you have any further questions or suggestions, please feel free to let us know. We welcome any thoughtful comments and constructive suggestions!
> >
> > Sincerely,
> >
> > Authors
> >
> > References
> >
> > [1] Tanishq Kumar, Zachary Ankner, Benjamin F Spector, Blake Bordelon, Niklas Muennighoff, Mansheej Paul, Cengiz Pehlevan, Christopher R´e, and Aditi Raghunathan. Scaling laws for precision. arXiv preprint arXiv:2411.04330, 2024. URL https://arxiv.org/abs/2411.04330.
> >
> > [2] Xiaoxuan Liu, Lanxiang Hu, Peter Bailis, Alvin Cheung, Zhijie Deng, Ion Stoica, and Hao Zhang. Online speculative decoding, 2024. URL https://arxiv.org/abs/2310.07177.

---

> > > ### Comment · Reviewer_FprC · 2024-11-26
> > >
> > > I am skeptical of your claim that this technique is a new quantization paradigm. From the perspective of applied quantization theory, there is nothing new here. You use existing methods (e.g., QuaRot and Atom) to represent weights and activations with low-precision integers, which is an established and well-studied paradigm.
> > >
> > > I interpret your proposal as a new quantized inference pipeline, which is fundamentally different than a new quantization paradigm. You offer interesting insights into the impact of quantization on multi-step reasoning and the impact of activation quantization on token prediction probabilities. However, with such a proposal, one must also consider the established paradigm of draft models in speculative decoding.
> > >
> > > As such, the benchmarking is still incomplete. The burden of proof is on the authors to identify if and when QSpec is more useful than the established paradigm. It is clear that it is useful from your analysis, but it is not clear if or when it is *more* useful than a specialized draft model. AI practitioners are already willing to pay the price of tuning a specialized draft model, so reducing (or in this case eliminating) that cost may have a limited impact.

---

> > > > ### Author Response · Authors · 2024-12-02
> > > > **Further response to reviewer (1/n)**
> > > >
> > > > Dear Reviewer FprC,
> > > >
> > > > We appreciate your interpretation of QSPEC and your concerns about benchmarking. In response, we have thoroughly compared QSPEC with EAGLE, a state-of-the-art specialized speculative decoding method, to demonstrate QSPEC's effectiveness.
> > > >
> > > > **Comparison with EAGLE**
> > > > - To comply with the reviewer's request, we conducted extensive experiments to compare EAGLE-Quant (using an FP16 draft model with W4A16 EAGLE), QSPEC, W4A16, and W4A4 on a single A100 GPU. We utilized the Llama-2-7b-chat-hf model and tested with batch sizes of 1, 8, and 16.
> > > > - For EAGLE-Quant, we opted to use an FP16 draft model combined with W4A16 EAGLE. This choice was made because the official EAGLE quantization (fast-gpt) does not support batching well. Additionally, when we attempted to quantize the EAGLE draft model using GPTQ, we observed a significant deterioration in the acceptance rate. We retained the FP16 draft model for EAGLE-Quant, which is the optimal case we observed, in our extended experiments.
> > > >
> > > > | Quantization | Batch Size | gsm8k (8-shot) | MATH (4-shot) | MBPP (0-shot) | HumanEval (0-shot) | ShareGPT | Lmsys-1k |
> > > > |--------------|------------|----------------|---------------|---------------|---------------------|----------|----------|
> > > > | **Eagle**    | 1          |  50.0         |   53.8        |   51.9        |   43.0              |  60.9    |  53.3    |
> > > > |              | 8          | 135.2         |  196.6        |  126.5        |  114.9              | 216.7    | 158.0    |
> > > > |              | 16         | oom           |   oom         |  134.0        |   94.8              | 231.3    |   oom    |
> > > > | **QSpec**    | 1          |  33.0         |   35.0        |   34.0        |   29.3              |  42.3    |  40.8    |
> > > > |              | 8          | 156.5         |  196.2        |  152.4        |  124.5              | 260.9    | 193.4    |
> > > > |              | 16         | 239.3         |  302.3        |  211.4        |  184.8              | 457.7    | 355.3    |
> > > > | **W4A16**    | 1          |  38.4         |   42.9        |   40.4        |   34.4              |  54.3    |  52.2    |
> > > > |              | 8          | 111.0         |  138.2        |  122.0        |  108.2              | 175.5    | 147.8    |
> > > > |              | 16         | 170.2         |  229.5        |  190.8        |  158.1              | 400.3    | 310.4    |
> > > > | **W4A4**     | 1          |  40.6         |   45.3        |   39.0        |   34.5              |  54.9    |  52.5    |
> > > > |              | 8          | 200.1         |  268.1        |  173.0        |  159.5              | 363.4    | 282.3    |
> > > > |              | 16         | 292.4         |  399.4        |  260.2        |  224.8              | 657.5    | 513.5    |
> > > >
> > > >
> > > >
> > > > - In previous table, EAGLE performs optimally with single-sequence inputs (batch size = 1) across all datasets, as noted in their paper. **However, its performance significantly deteriorates with larger batch sizes (e.g., 8 or 16)**. This is primarily because EAGLE's high acceptance rate relies heavily on a tree structure whose size scales with the batch size. The necessity of this tree structure arises from the fact that EAGLE's draft model is largely built around an additional decode block specifically trained for drafting which lacks high consistency with the target model, making the tree structure essential for bridging discrepancies and maintaining alignment. As the batch size increases, the computation load on the target model increases due to the scaled tree structure. The computation on the target model shifts from being memory-bound to compute-bound, leading to performance degradation under W4A16.
> > > > - Furthermore, as we mentioned in our paper, speculative methods like EAGLE can **face out-of-memory (OOM) issues**—for instance, we observed that EAGLE encounters OOM at a batch size of 16. This is because the key-value (KV) storage requirements of EAGLE's draft model grow substantially with larger batches, causing OOM problems.
> > > > - In contrast, as noted in our previous response, QSPEC is both more batch-friendly and memory-efficient. In QSPEC's drafting phase (W4A4), **QSPEC does not require a tree structure to maintain a high acceptance rate**, so the workload does not scale with tree size as the batch size increases. Additionally, QSPEC does not introduce any extra memory requirements compared to the high-precision method, as we share the weights and KV cache with W4A16. As a result, QSPEC demonstrates better performance at larger batch sizes (8 and 16) compared to EAGLE, but also doesn't have the OOM issue.

---

> > > > > ### Comment · Reviewer_FprC · 2024-12-02
> > > > >
> > > > > Thank you for the responses. I am happy to raise my score.
> > > > >
> > > > > If you are able to, please include these results in the appendix.

---

> > > > > > ### Author Response · Authors · 2024-12-04
> > > > > > **Further response to reviewer (2/n)**
> > > > > >
> > > > > > Dear Reviewer FprC,
> > > > > >
> > > > > > Thank you for your feedback and for deciding to raise the score. Your suggestions have been very helpful in improving our work, and we will manage to include these results in the appendix as you suggested.
> > > > > >
> > > > > > Sincerely,
> > > > > >
> > > > > > Authors

---

### Official Review · Reviewer_9onb · 2024-10-27

**Soundness:** 2
**Presentation:** 3
**Contribution:** 2
**Rating:** 6
**Confidence:** 3

**Summary:**

The authors propose a weight-sharing speculative decoding framework for quantized models with varying activation precisions. The core insight of this work is that aggressive weight-activation quantization methods, such as W4A4, often result in performance degradation on complex multi-step reasoning tasks. To address this performance gap while retaining the efficiency of low bit-width models, the framework employs W4A4 to draft the output and W4A16 to verify it, sharing 4-bit weights but producing outputs at different precisions. The KV-cache is subsequently updated by the high-precision model, enhancing the framework with a high-quality KV-cache. Experiments demonstrate that the framework achieves comparable performance to full W4A16 models, while delivering higher throughput with W4A4-based drafting.

**Strengths:**

The paper presents an interesting observation that claims of "lossless" performance in many quantization studies are often skewed by limited evaluation benchmarks, which overlook the negative impacts of activation quantization. To address this issue, the authors propose a weight-sharing mixed-precision speculative decoding framework. Unlike conventional heterogeneous speculative decoding, which uses a small model for drafting and a significantly larger model for verification, the authors argue that homogeneous speculative decoding with weight-sharing is more suitable for edge scenarios. Although the framework may appear similar to conventional speculative decoding, experiments demonstrate its advantages in both accuracy and throughput.

**Weaknesses:**

While the observation and intuition behind this work are compelling, some experiments seem to be missing, which weakens the support for the authors' claims. For instance, Tables 1 and 3 lack comparisons with FP16 models. Demonstrating the performance differences between FP16 and W4A16 on complex multi-step reasoning tasks would help the community better understand the true performance gaps between FP16 models and their quantized counterparts. Additionally, Table 4 lacks throughput evaluations for W4A4. Including a latency and accuracy trade-off comparison between the framework and FP16, W4A16, and W4A4 models would provide valuable insights. The absence of these experiments may mislead readers and hinder a full assessment of the proposed framework. I am willing to raise my score if authors address aforementioned issues.

**Questions:**

- Could the authors provide the accuracy of the FP16 models in Tables 1 and 3?
- Could the authors provide the throughput of the FP16 and W4A4 models in Table 4 as well as in Figures 4 and 5?
- Could the authors include a figure illustrating the throughput and accuracy trade-offs for the proposed framework compared to all baselines?

---

> ### Author Response · Authors · 2024-11-21
> **Response to reviewer (1/n)**
>
> Dear Reviewer 9onb,
>
> We sincerely thank you for your understanding of our work and constructive suggestions! In response to your questions, we have conducted additional experiments for supplementary results. Hope that they could address your concerns!
>
> Question 1: Accuracy of FP16 models in Tables 1 and 3.
> - Actually, we have provided the results of FP16 natively in Table 1, but name it as W16A16 to highlight the differences from W4A16 and W4A4 methods. Sorry for the confusion caused by this misleading notation! We have clarified it in our updated submission. Additionally, we have added the accuracy of W16A16 (i.e., FP16) to Table 3. You can now check it for details.
>
> Question 2: Throughput of FP16 and W4A4 models.
> - Due to a cooling system failure, our A100 server is under repair. Hence, for comprehensive clarifications, we repeat our core experiments and supplement the requested analysis on currently available L20 GPUs. All the new results are organized in the appendix of our latest submission. Correspondingly, Table 4, Figure 4, and Figure 5 are reproduced in Table 5, Figure 7, and Figure 8 with the supplementary throughput of the FP16 (i.e., W16A16) and W4A4 models.
>
> Question 3: Throughput and accuracy comparison of all the methods.
> - For your reference, we add Figure 6 in the Appendix at the end of Page 15. This figure illustrates the comparison of accuracy and efficiency among W16A16, W4A16, W4A4, and QSpec across various datasets with batch sizes of 8 and 16, respectively. Specifically, W16A16 exhibits the best performance with the highest precision. In contrast,  W4A4 shows the worst even with a quality loss exceeding 30%, resulting in its infeasibility in practice. Between them, W4A16 and QSpec perform nearly identically, both slightly below W16A16 due to weight quantization for memory saving. In terms of efficiency, W4A4 leverages low-bit computation cores, and achieves the highest throughput. Compared to W4A16 and W16A16, QSpec provides more efficient inference, attributed to its rapid drafting capabilities and parallel verification.
>
> Hope that these supplementary results could provide a clearer understanding of our method, and you can reconsider our scores accordingly! If you have any further questions, please feel free to let us know, and we welcome any insightful discussions!
>
> Sincerely,
>
> Authors

---

> ### Comment · Reviewer_9onb · 2024-11-22
>
> Thank you for the reply. The authors have addressed my concerns in the revision, which has led me to raise my score.
>
> In summary, I believe highlighting the limitations of current low bit-width quantization methods, which are skewed by narrow evaluation benchmarks and lack generalizability, is a valuable contribution. Although the proposed method appears to be a relatively straightforward solution by adopting existing W4A4 and W4A16 techniques from prior work and integrating them with a speculative decoding approach, I do appreciate the technical efforts involved and the comprehensive experiments that demonstrate improvements in both throughput and generalizability.
>
> I would encourage the authors to consider using LLM-QBench [1] for larger-scale benchmarking and to report their method's performance against other low bit-width baselines on LLM-QBench.
>
> [1] Gong, Ruihao, et al. "LLM-QBench: A Benchmark Towards the Best Practice for Post-training Quantization of Large Language Models." arXiv preprint arXiv:2405.06001 (2024).

---

> > ### Author Response · Authors · 2024-11-26
> > **Response to reviewer (2/n)**
> >
> > Dear Reviewer 9onb,
> >
> > We sincerely appreciate your thoughtful comments, your acknowledgment of the contributions in our work, and your reconsideration of our score.
> >
> > Your suggestion to use LLM-QBench for larger-scale benchmarking is valuable, and we will consider it seriously in future iterations of our work, which will help us find deeper insights into the performance of QSpec.
> >
> > Once again, we are grateful for your constructive feedback and encouraging remarks, which have significantly contributed to improving our work.
> >
> > Sincerely,
> >
> > Authors

---

### Official Review · Reviewer_mqt3 · 2024-11-02

**Soundness:** 2
**Presentation:** 3
**Contribution:** 2
**Rating:** 5
**Confidence:** 4

**Summary:**

The paper brings an elegant idea to use faster 4b quantized activations for drafting, and use full 16b activations for decoding.

There are significant weaknesses in completeness of the analysis, specifically lack of comparison with other quantization methods in terms of speedup.

**Strengths:**

It is an elegant idea to use faster 4b quantized activations for drafting, and use full 16b activations for decoding - it saves the memory for weights and helps to align draft / target distributions.

**Weaknesses:**

1) The paper does not reported acceptance rates

2) The paper no comparison to other lossless speculative decoding methods, namely Eagle2, Medusa2 with top speedups, BiTA with negligible VRAM overhead.

3) table 1: WikiText-2 PPL # 8.70 7.87 (+9.20%) 8.58 (+1.15%) - why ppl is lower for W4 methods?

4) table 4: 7B W4A16 HumanEval seems too fast (300+ vs other methods)

5) the paper fails to notice that W4A4 is more memory efficient - need to compare using same memory budget

6) the abstract should mention average improvement, not peak

7) Since this is a lossless method, tests should mostly focus on speed, not on quality. So please add speedup comparisons vs other quantization methods, not just to W4A16 baseline.

8) Llama family should refer to L2 paper, not just L3

9) Table4 - add model names

**Questions:**

1) How would your method compare in speedup to other lossless speculative decoding methods, namely Eagle2, Medusa2 (with top speedups), BiTA (with negligible VRAM overhead).

2) How would your method performs with tree-based speculation?

3) Is the method good for batched generation? What is its performance vs batch size?

4) please report your observed acceptance rates

---

> ### Author Response · Authors · 2024-11-23
> **Response to reviewer (1/n)**
>
> Dear Reviewer mqt3,
>
> Thank you so much for your thorough reviews and insightful comments on our work! We really appreciate the time and effort you have devoted to our submission. Below, we first discuss the extra contribution and potential impact of QSpec from both academic and industrial perspectives, and then respond to your comments one by one. Hope that they could fully address your concerns!
>
>
> **Contribution on the analysis of existing quantization methods.**
>
> - To the best of our knowledge, we are **the first to highlight the significant deficiencies of W4A4 quantization methods on multi-step reasoning tasks**. Previously, all W4A4 quantization studies claimed that their methods achieved competitive performance to a FP16 solution, encouraging users to directly apply their solutions across all tasks. This misrepresentation stems from biased evaluation protocols. We point out this issue, informing users of substantial performance losses which they may suffer with W4A4 methods. Our findings will assist users in selecting appropriate quantization methods, and encourage researchers to incorporate multi-step reasoning tasks into evaluation protocols for a more comprehensive assessment.
>
> **Potential impact of QSpec.**
>   - **From the academic perspective.**
>     - **QSpec validates the feasibility of nearly cost-free switching** between two quantization schemes of a shared weight-quantized model, as well as their high token-level similarities. This may directly illuminate the future development of quantization schemes.
>     - **QSpec decouples the speed enhancement and quality preservation in existing quantization schemes**. Previously, researchers had to devote their efforts to balance performance loss against acceleration. Now, QSpec assigns the roles of speed improvement and performance assurance to the draft model and verify model, respectively. This separation shifts researchers' focus from the challenging balance to a concentrated effort on acceleration. This new perspective unlocks significant opportunities for further research.
>
>   - **From the industrial perspective.**
>     - **QSpec prompts chip vendors to reconsider their design**. Due to the compromised performance of low-precision activation quantization, **many modern NPUs (e.g., Ascend and GraphCore) primarily utilize FP16/BF16 computation cores, resulting in the underutilization of low-precision kernels**. However, QSpec offers a general plug-and-play solution that harnesses low-precision architectures to enhance efficiency without sacrificing quality. Notably, we have received feedback from one chip vendor, indicating that QSpec has inspired them to rethink the proposition of investment that they should allocate to low-bit cores and instructions.
>
> BTW, a recent paper titled "Scaling Laws for Precision" demonstrates that performance degradation becomes increasingly severe as the number of quantization bits decreases, particularly in activation quantization [1]. This finding undoubtedly hampers the development of further quantization schemes. However, as mentioned above, QSpec decouples speed enhancement and quality preservation in quantization methods, focusing quantization efforts on acceleration rather than the tough balance against performance loss. This may foster the continuous development of quantization methods.

---

> ### Author Response · Authors · 2024-11-23
> **Response to reviewer (2/n)**
>
> **Weakness 1: Reported acceptance rates**
>
> - As shown in Figure 5 of our original submission, we visualize the acceptance rates of Llama3.2-3b and Llama3-8b-instruct models with respect to the draft token length γ. Even with γ=7, **the token acceptance rate of QSpec remains high** at approximately 70%, compared to 28–58% observed on a 160m-7b draft-target model pair with γ=5 in conventional speculative decoding [2]. Additionally, we test the acceptance rates of Atom-based and Quarot-based QSpec across diverse datasets. As detailed in the table below, **the consistently high acceptance rates demonstrate the robustness of QSpec across both quantization methods and benchmarks.** Thanks for your suggestions! These results will be further supplemented into our final paper.
> | Quantization | lmsys | sharegpt | gsm8k | MATH  | MBPP  | HumanEval |
> |--------------|-------|----------|-------|-------|-------|-----------|
> | Atom         | 0.867 | 0.838    | 0.915 | 0.894 | 0.886 | 0.914     |
> | QuaRot       | 0.830 | 0.816    | 0.890 | 0.889 | 0.854 | 0.882     |
>
> **Weakness 2: Comparison with other speculative methods.**
> - As highlighted in our Introduction section and echoed by Reviewer 9onb, "the authors propose a weight-sharing speculative decoding framework for quantized models with varying activation precisions." Instead of integrating quantization into speculative decoding, **QSpec is more like a new quantization paradigm**, which leverages the spirit of speculative decoding to solve the problems met in current quantization schemes.
> - As mentioned in the potential impacts, **QSpec introduces indispensable advantages for quantization methods (e.g., efficiency-performance decoupling) and hardware design**, differentiating it from regular dual-model speculative decoding.
> - Typically, quantization methods are deployed in environments (e.g., edge devices) with severely limited resources, particularly regarding memory. This eliminates almost all existing speculative decoding, due to increased weight and KV cache consumption. In contrast, the design of QSpec, including both weight-sharing and KV cache overwriting, circumvents the extra memory overheads over W4A16, **aligning with the quantization scenarios**. Together with the above analysis, these advantages make comparisons with regular memory-extensive speculative decoding approaches not that necessary, due to their deviation from the initial motivation of our work.
> - **QSpec is completely orthogonal to existing speculative decoding methods**. Specifically, QSpec can seamlessly integrate with existing speculative schemes using a hierarchical structure, such as "(Draft model ➔ W4A4) ➔ W4A16" or "(W4A4 ➔ W4A16) ➔ Verify model". This integration will provide additional acceleration benefits for both QSpec and existing solutions.
> - **QSpec is plug-and-play, requiring no prerequisites or complex operations.** In contrast, all of Medusa, EAGLE and BiTA necessitate retraining.
> - None of Medusa, EAGLE and BiTA support large batch sizes, whereas **QSpec is hardware-friendly and fully accommodates batched serving**. As shown in Figure 22 of Medusa [3] and Table 7 of EAGLE [4], the speedup ratios for both methods decrease with the increase of batch size. Notably, for batch sizes larger than 16, Medusa slows down the generation process, demonstrating poor support for large batch sizes. EAGLE does not report performance for batch sizes larger than 4, while BiTA only conducts analysis with the batch size of 1. We will also include this discussion in our final paper.
>
> **Weaknesses 3 & 4: Data anomaly in Table 1 and Table 4**
>
> We sincerely apologize for the errors in the reported data. The PPL anomaly in Table 1 stems from an early pre-experiment, which is neglected for modification. For Table 4, the discrepancy is caused by a clerical error. Both mistakes do not challenge our existing conclusions. Thanks for your careful reviews, and we have been motivated to thoroughly recheck all the data in our submission to ensure its correctness.

---

> ### Author Response · Authors · 2024-11-23
> **Response to reviewer (3/n)**
>
> **Weakness 5: Comparison with W4A4 using the same memory budget**
>
> 1. As shown in our analysis of existing quantization methods on multi-step reasoning tasks, **W4A4 shows the worst performance, even with a quality loss exceeding 30%** when compared to W4A16, resulting in its infeasibility in practice.
> 2. For your reference, we **supplement Figure 6 to illustrate the comparison of accuracy and efficiency among W16A16, W4A16, W4A4, and QSpec** across various datasets with batch sizes of 8 and 16, respectively. Hope that it could help you further understand the performance of various methods.
> 3. The memory consumption mainly consists of **weights, KV cache, and activations**, which are **not directly related to the computation cores/precision**. Specifically, activations are released immediately after matrix operations, only occupying memory temporarily, while weights and KV cache are shared between W4A4 and W4A16 in QSpec. Despite slightly more memory consumption than W4A4, we do believe that this **deserves a 30% performance gain**.
> 4. Finally, we tried to find good solutions to control the same memory budget for the comparison between QSpec and W4A4, but failed. If you have any suggestions, please feel free to let us know!
>
> **Weakness 6: Average Improvement in Abstract**
>
> Thank you for this feedback! We apologize for the oversight of reporting the average improvements in the abstract, although they have been detailed in Table 4. We will follow your suggestions, and revise the abstract to reflect the average improvements in our final version.
>
> **Weakness 7: Comparison of speedup & Comparison with other quantization methods**
>
> **1. Comparison of speedup.**
> - Since the initial motivation of QSpec is to alleviate the performance-efficiency trade-off of quantization methods (i.e., W4A4 offers speed enhancements over W4A16, but suffers significant performance degradation), we validate the preserved performance of QSpec in Table 3, while **presenting its acceleration over W4A16 in Table 4**. We feel so sorry for the potentially misleading presentation!
> - Besides, in the rebuttal process, we further extend our experiments on NVIDIA L20 GPUs, and **supplement additional analysis, including extra comparisons of efficiency across all the methods**. If interested, please refer to the updated submission for details!
>
> **2. Comparison with other quantization methods.**
> - Firstly, as mentioned above, existing W4A4 methods generally suffer excessively inferior performance.
> - Besides, **W4A4 and W4A16 are not restricted to specific quantization methods, nor is QSpec.** As exemplied by Atom and QuaRot in our paper, any W4A4 method can be seemingly integrated into QSpec with the plug-and-play advantage, and achieves consistent performance and enhanced efficiency compared to its W4A16 counterpart.
>
> **Weaknesses 8 & 9: Reference and Table Corrections**
>
> - Thank you so much for bringing these details to our attention! We appreciate your concrete suggestions, and will make the revisions accordingly.
>
>
>
> **Question 1: Comparison with other speculative methods.**
>
> - We elaborate this explanation in the clarification on Weekness 2. Please refer to it for details.
>
> **Question2: How would your method perform with tree-based speculation?**
> - Based on our understanding, tree-based speculative decoding first drafts tree-like candidate tokens, before matching the rightest sub-path with the verify model [5]. This design mainly involves the  draft and verify pattern, and **should be orthogonal to our method**. Intuitively, **our draft model (i.e., W4A4 with shared weights) can also speculate tree-like candidate tokens**, and utilizes the verify model (i.e., W4A16 with shared weights) to match the most correct sub-path. This can **further enhance the performance of QSpec**. Exploring such integrations in our scenario could be a valuable direction for future work, and we will also include this discussion in our final paper.
> - As for the reason why we do not combine it into QSpec, we want to focus QSpec on its core advantage of two complementary quantization schemes with shared weights and KV cache, while keeping it compatible with regular optimization techniques developed for speculative decoding, rather than limiting it to one specific optimization.

---

> ### Author Response · Authors · 2024-11-23
> **Response to reviewer (4/n)**
>
> **Question3: Suitability for batched generation**
>
> Thanks for raising this insightful question!
>
> - **Due to the hardware-friendly property, QSpec is inherently suitable for batched generation**, and exhibits robust throughput enhancement across a range of batch sizes, as shown in Figures 4 and 5, as well as in Figures 6-8 of our supplementary analysis.
> - In contrast, **both Medusa and EAGLE primarily operate with the batch size as 1**, and suffer remarkable performance deterioration at larger batch sizes, as shown in Figure 22 of Medusa [3] and Table 7 of EAGLE [4]. Notably, for batch sizes larger than 16, Medusa even slows down, instead of accelerating, the generation process, demonstrating its poor support for large batch sizes. EAGLE does not report performance for batch sizes larger than 4. In contrast, QSpec is plug-and-play, and does not require structural modifications or retraining, allowing it to perform well even in these settings.
>
> **Question4: Reported acceptance rates.**
>
> - Please refer to our explanation to Weakness 1 for details.
>
> Hope that our responses can fully address your concerns, and help you further understand the contribution and potential impact of our work! If you have any further questions, please feel free to let us know! We welcome any valuable discussions, and, if possible, your reconsideration of our scores!
>
> Sincerely,
>
> Authors
>
>
> **References**
>
> [1] Tanishq Kumar, Zachary Ankner, Benjamin F Spector, Blake Bordelon, Niklas Muennighoff, Mansheej Paul, Cengiz Pehlevan, Christopher R´e, and Aditi Raghunathan. Scaling laws for precision. arXiv preprint arXiv:2411.04330, 2024. URL https://arxiv.org/abs/2411.04330.
>
> [2] Xiaoxuan Liu, Lanxiang Hu, Peter Bailis, Alvin Cheung, Zhijie Deng, Ion Stoica, and Hao Zhang. Online speculative decoding, 2024. URL https://arxiv.org/abs/2310.07177.
>
> [3] Tianle Cai, Yuhong Li, Zhengyang Geng, Hongwu Peng, Jason D. Lee, Deming Chen, and Tri Dao. Medusa: Simple llm inference acceleration framework with multiple decoding heads, 2024. URL https://arxiv.org/abs/2401.10774.
>
> [4] Yuhui Li, Fangyun Wei, Chao Zhang, and Hongyang Zhang. Eagle: Speculative sampling requires rethinking feature uncertainty, 2024. URL https://arxiv.org/abs/2401.15077.
>
> [5] Xupeng Miao, Gabriele Oliaro, Zhihao Zhang, Xinhao Cheng, Zeyu Wang, Zhengxin Zhang, Rae Ying Yee Wong, Alan Zhu, Lijie Yang, Xiaoxiang Shi, Chunan Shi, Zhuoming Chen, Daiyaan Arfeen, Reyna Abhyankar, and Zhihao Jia. Specinfer: Accelerating large language model serving with tree-based speculative inference and verification. In Proceedings of the 29th ACM International Conference on Architectural Support for Programming Languages and Operating Systems, Volume 3, ASPLOS ’24. ACM, April 2024. doi: 10.1145/3620666.3651335. URL http://dx.doi.org/10.1145/3620666.3651335.

---

> > ### Comment · Reviewer_mqt3 · 2024-11-25
> > **remaining concerns**
> >
> > Dear Authors, thank you for the extensive rebuttal. These are my major remaining areas of concern:
> > - Comparison with W4A4 using the same memory budget would still be welcome here. One way to do that is to chart two frontiers for w4a4 and w4a16 in the memory / quality axes. This kind of comparison is quite common in quantization papers.
> > - your willingness to correct the omissions in the paper is encouraging. However the amount of corrections and additons is a bit worrying
> > - proper comparison with EAgle and other methods remains needed here. You give some qualitative discussion in this area, but a chart or table of speedup vs batch size vs quality would be worth many words.

---

> ### Author Response · Authors · 2024-12-02
> **Further response to reviewer (1/n)**
>
> Dear Reviewer mqt3,
>
> Thank you for pointing the remaining concerns. We'd like to address them by points.
>
> **Concern 1: Chart two frontiers for w4a4 and w4a16 in the memory / quality axes**
> - We collected memory and quality metrics by conducting experiments under both W4A4 and QSPEC settings on a single A100 GPU (40GB). Specifically, we used the MBPP dataset with the LLaMA2-7B model and the GSM8K dataset with the LLaMA2-13B model. As the reviewer anticipated, W4A4 is more memory-efficient but yields much poorer quality compared to QSPEC. Note that this difference stems entirely from the precision of KV. QSPEC can potentially use the KV4 method to achieve the same memory consumption as shown in W4A4 KV4.
>
> | Quantization   | Setting    | 8       | 64       | 128      | 256      | Max Batch Size | Quality (Accuray) |
> |----------------|------------|---------|----------|----------|----------|----------------|----------------|
> | QSpec KV16     | Llama2-7B  | 6282    | 18434    | 31690    | -        | oom@166        | 14.29%         |
> |                | Llama2-13B | 9938    | 26729    | -        | -        | oom@104        | 32.00%         |
> | W4A4 KV4       | Llama2-7B  | 5366    | 9806     | 14940    | 24480    | oom@490        | 7.67%          |
> |                | Llama2-13B | 8656    | 14882    | 21550    | 35420    | oom@296        | 25.00%         |
>
> **Concern 2: Amount of corrections and additons**
> - With the modifications we have implemented, **our performance has improved while our original claims remain unchanged.** Although we did not include the details of our optimized attention kernel in the manuscript, it plays a crucial role in QSPEC's performance. Recently, we have enhanced attention kernel implementation, further boosting QSPEC's overall performance.
> - Regarding the extended L20 results, as we previously mentioned, we had to temporarily switch to different hardware due to server maintenance. **However, this change has helped demonstrate the generalizability of our solution across different hardware platforms.**

---

> > ### Author Response · Authors · 2024-12-02
> > **Further response to reviewer (2/n)**
> >
> > **Concern 3: Comparison with EAGLE**
> > - To comply with the reviewer's request, we conducted extensive experiments to compare EAGLE-Quant (using an FP16 draft model with W4A16 EAGLE), QSPEC, W4A16, and W4A4 on a single A100 GPU. We utilized the Llama-2-7b-chat-hf model and tested with batch sizes of 1, 8, and 16.
> > - For EAGLE-Quant, we opted to use an FP16 draft model combined with W4A16 EAGLE. This choice was made because the official EAGLE quantization (fast-gpt) does not support batching well. Additionally, when we attempted to quantize the EAGLE draft model using GPTQ, we observed a significant deterioration in the acceptance rate. We retained the FP16 draft model for EAGLE-Quant, which is the optimal case we observed, in our extended experiments.
> >
> > | Quantization | Batch Size | gsm8k (8-shot) | MATH (4-shot) | MBPP (0-shot) | HumanEval (0-shot) | ShareGPT | Lmsys-1k |
> > |--------------|------------|----------------|---------------|---------------|---------------------|----------|----------|
> > | **Eagle**    | 1          |  50.0         |   53.8        |   51.9        |   43.0              |  60.9    |  53.3    |
> > |              | 8          | 135.2         |  196.6        |  126.5        |  114.9              | 216.7    | 158.0    |
> > |              | 16         | oom           |   oom         |  134.0        |   94.8              | 231.3    |   oom    |
> > | **QSpec**    | 1          |  33.0         |   35.0        |   34.0        |   29.3              |  42.3    |  40.8    |
> > |              | 8          | 156.5         |  196.2        |  152.4        |  124.5              | 260.9    | 193.4    |
> > |              | 16         | 239.3         |  302.3        |  211.4        |  184.8              | 457.7    | 355.3    |
> > | **W4A16**    | 1          |  38.4         |   42.9        |   40.4        |   34.4              |  54.3    |  52.2    |
> > |              | 8          | 111.0         |  138.2        |  122.0        |  108.2              | 175.5    | 147.8    |
> > |              | 16         | 170.2         |  229.5        |  190.8        |  158.1              | 400.3    | 310.4    |
> > | **W4A4**     | 1          |  40.6         |   45.3        |   39.0        |   34.5              |  54.9    |  52.5    |
> > |              | 8          | 200.1         |  268.1        |  173.0        |  159.5              | 363.4    | 282.3    |
> > |              | 16         | 292.4         |  399.4        |  260.2        |  224.8              | 657.5    | 513.5    |
> >
> >
> >
> > - In previous table, EAGLE performs optimally with single-sequence inputs (batch size = 1) across all datasets, as noted in their paper. **However, its performance significantly deteriorates with larger batch sizes (e.g., 8 or 16)**. This is primarily because EAGLE's high acceptance rate relies heavily on a tree structure whose size scales with the batch size. The necessity of this tree structure arises from the fact that EAGLE's draft model is largely built around an additional decode block specifically trained for drafting which lacks high consistency with the target model, making the tree structure essential for bridging discrepancies and maintaining alignment. As the batch size increases, the computation load on the target model increases due to the scaled tree structure. The computation on the target model shifts from being memory-bound to compute-bound, leading to performance degradation under W4A16.
> > - Furthermore, as we mentioned in our paper, speculative methods like EAGLE can **face out-of-memory (OOM) issues**—for instance, we observed that EAGLE encounters OOM at a batch size of 16. This is because the key-value (KV) storage requirements of EAGLE's draft model grow substantially with larger batches, causing OOM problems.
> > - In contrast, as noted in our previous response, QSPEC is both more batch-friendly and memory-efficient. In QSPEC's drafting phase (W4A4), **QSPEC does not require a tree structure to maintain a high acceptance rate**, so the workload does not scale with tree size as the batch size increases. Additionally, QSPEC does not introduce any extra memory requirements compared to the high-precision method, as we share the weights and KV cache with W4A16. As a result, QSPEC demonstrates better performance at larger batch sizes (8 and 16) compared to EAGLE, but also doesn't have the OOM issue.

---

> > > ### Author Response · Authors · 2024-12-04
> > > **Further response to reviewer (3/n)**
> > >
> > > Dear Reviewer mqt3,
> > >
> > > We fully understand that you may have been very busy recently!
> > >
> > > Given that the rebuttal deadline is approaching, we genuinely hope that our clarifications have addressed your concerns and will encourage you to reconsider the scores. If you have any further questions, please do not hesitate to discuss them with us！
> > >
> > > Looking forward to your further feedback!
> > >
> > > Sincerely,
> > >
> > > Authors

---

### Official Review · Reviewer_PcTG · 2024-11-03

**Soundness:** 3
**Presentation:** 3
**Contribution:** 2
**Rating:** 6
**Confidence:** 5

**Summary:**

This paper introduces a new method called QSPEC, which seamlessly integrates two complementary quantization schemes for speculative decoding. QSPEC utilizes W4A4 quantized LLMs to draft tokens and verifies these drafted tokens with W4A16 quantized LLMs, achieving results comparable to those of W4A16 quantized LLMs in generation tasks.

**Strengths:**

1. The authors highlight that quantized models perform worse on generation tasks (e.g., code and math) compared to multiple-choice tasks, particularly when using 4-bit quantized activations. This is a significant issue that the community should address.

2. To my knowledge, this paper is the first to combine quantization and speculative decoding. The authors analyze their compatibility, focusing on token prediction similarity, acceptance rate, shared weights, and key-value caches.

**Weaknesses:**

1. The novelty of this paper is limited. While the combination of speculative decoding and quantization is an interesting topic, the authors do not introduce any new methods. I suggest that the authors explore ways to enhance quantization methods in speculative decoding. For instance, do different quantization techniques, such as Atom and QuaRot, result in varying acceptance rates?

2. It is recommended to make more comparisons with previous speculative decoding methods in order to better demonstrate the benefits of integrating quantization into speculative decoding in Table 4. Due to the shared weights and key - value cache, the memory consumption might be one of the advantages of this method, and this aspect can also be added to Table 4.

3. The speedup ratio compared with previous speculative decoding methods such as Medusa and EAGLE is weak.

**Questions:**

Why does the speedup ratio relative to W4A16 decline as the batch size increases, as depicted in Table 4? I'm intrigued by the overall inference latency difference between the W4A16 and W4A4 models. Could you provide the latency of W4A4 and W4A16 at both the generation and prefilling stages?

---

> ### Author Response · Authors · 2024-11-21
> **Response to reviewer (1/n)**
>
> Dear Reviewer PcTG,
>
> Thank you so much for your great efforts and constructive comments on our work! We have read your feedback carefully and made the following clarifications. Hope that they could further address your concerns!
>
>
>
>
> **Weakness 1: Lack of novelty.**
>
> To relieve this concern, we want to discuss the importance of our contribution to the analysis of existing quantization methods before discussing the design and potential impact of QSpec on future developments from both academic and industrial perspectives.
>
> **1. Contribution to the analysis of existing quantization methods.**
> - To the best of our knowledge, we are **the first to highlight the significant deficiencies of W4A4 quantization methods on multi-step reasoning tasks**. Previously, all W4A4 quantization studies claimed that their methods achieved competitive performance to an FP16 solution, encouraging users to directly apply their solutions across all tasks. This misrepresentation stems from biased evaluation protocols. We point out this issue, informing users of substantial performance losses that they may suffer with W4A4 methods. Our findings will assist users in selecting appropriate quantization methods, and encourage researchers to incorporate multi-step reasoning tasks into evaluation protocols for a more comprehensive assessment.
>
> **2. Design of QSpec.**
> - The initial motivation of QSpec is to alleviate the performance-efficiency trade-off of quantization methods (i.e., W4A4 offers speed enhancements over W4A16, but suffers significant performance degradation). Besides, we expect our method to avoid any extra overhead compared to W4A16, since deployment environments for quantization methods typically have severely limited resources, particularly in terms of memory. To fulfill these requirements, our first design is weight sharing, effectively saving the model parameters by half. This sharing also facilitates our second design named "KV cache overwriting", which relies on weight sharing as a prerequisite. Despite the simplicity of QSpec (actually, we regard this as an advantage for easier deployment and robust performance), we ingeniously **circumvent additional overhead compared to W4A16, and effectively boost its efficiency without performance sacrifices**.
>
> **3. Potential academic impact of QSpec.**
> - **QSpec validates the feasibility of nearly cost-free switching** between two quantization schemes
> of a shared weight-quantized model, as well as their high token-level similarities. This may directly illuminate
> the future development of quantization schemes.
> - **QSpec decouples the speed enhancement and quality preservation in existing quantization schemes.** Previously, researchers had to devote their efforts to balancing performance loss against acceleration. Now, QSpec assigns the roles of speed improvement and performance assurance to the draft model and verify model, respectively. This separation shifts researchers' focus from the challenging balance to a concentrated effort on acceleration. This new perspective unlocks significant opportunities for further research.
>
> **4. Potential industrial impact of QSpec.**
> - **QSpec prompts chip vendors to reconsider their design.** Due to the compromised performance of low-precision activation quantization, **many modern NPUs (e.g., Ascend and GraphCore) primarily utilize FP16/BF16 computation cores, resulting in the underutilization of low-precision kernels.** However, QSpec offers a general plug-and-play solution that harnesses low-precision architectures to enhance efficiency without sacrificing quality. Notably, we have received feedback from one chip vendor, indicating that QSpec has inspired them to rethink the proposition of investment that they should allocate to low-bit cores and instructions.
>
> BTW, a recent paper titled "Scaling Laws for Precision" demonstrates that performance degradation becomes increasingly severe as the number of quantization bits decreases, particularly in activation quantization [1]. This finding undoubtedly hampers the development of further quantization schemes. However, as mentioned above, QSpec decouples speed enhancement and quality preservation in quantization methods, focusing quantization efforts on acceleration rather than the tough balance against performance loss.

---

> ### Author Response · Authors · 2024-11-21
> **Response to reviewer (2/n)**
>
> **Weakness 2: Comparison with other speculative methods.**
> - As mentioned by Reviewer 9onb, "the authors propose a weight-sharing speculative decoding framework for quantized models with varying activation precisions." Instead of integrating quantization into speculative decoding, we'd like to regard **QSpec as a new quantization paradigm**. As indicated in our Introduction section, these two perspectives are different. Typically, quantization methods are deployed in environments (e.g., edge devices) with severely limited resources, particularly regarding memory. This eliminates almost all existing speculative decoding, due to significantly increased weight and KV cache consumption.
> - **QSpec is completely orthogonal to existing speculative decoding methods.** Thanks to the plug-and-play advantage, QSpec can seamlessly integrate with existing speculative schemes using a hierarchical structure, such as "(Draft model ➔ W4A4) ➔ W4A16" or "(W4A4 ➔ W4A16) ➔ Verify model." This integration will provide additional acceleration benefits for both QSpec and existing solutions.
> - As mentioned in the clarification on weakness 1, **QSpec introduces indispensable advantages for quantization methods (e.g., efficiency-performance decoupling) and hardware design.**
> - For memory consumption, we exhibit the comparison of individual quantization schemes, regular speculative decoding, and QSpec in Table 2 of our submission.
>
> **Weakness 3: Comparison to Medusa and EAGLE**
> - In Weakness 2, we clarify **the indispensable advantages of QSpec for quantization methods, its suitability for memory-constrained scenarios, its impact on hardware design, and its orthogonality to existing speculative decoding methods**.
> - Besides, **both Medusa and EAGLE necessitate retraining**. In contrast, our approach is **plug-and-play, requiring no prerequisites or complex operations.**
> - **Neither Medusa nor EAGLE supports large batch sizes, whereas our method is hardware-native and fully accommodates batched serving.** As shown in Figure 22 of Medusa [2] and Table 7 of EAGLE [3], the speedup ratios for both methods decrease with the increase of batch size. Notably, for batch sizes larger than 16, Medusa slows down the generation process, demonstrating poor support for large batch sizes. EAGLE does not report performance for batch sizes larger than 4. In contrast, QSpec is plug-and-play, and does not require structural modifications or retraining, allowing it to perform well even in these settings.

---

> ### Author Response · Authors · 2024-11-21
> **Response to reviewer (3/n)**
>
> **Question 1: Declined speedup ratio with the increase of batch size.**
> - According to the paper named "MagicDec: Breaking the Latency-Throughput Tradeoff for Long Context Generation with Speculative Decoding" [4], when sequence lengths are relatively short (e.g., less than 1024 in MagicDec's setting), **the verification stage becomes more computationally expensive with the increase of batch sizes**. This might bound the throughput at the beginning of the generation, where prefilling lengths are generally less than 1024 across regular academic benchmarks. As mentioned earlier, both Figure 22 of Medusa [2] and Table 7 of EAGLE [3] report similar phenomena that the speedup ratios decrease as batch size increases.
>
> **Question 2: Latency of W4A4 and W4A16 at both generation and prefilling stages.**
> - Due to a cooling system failure, our A100 server is under repair. Hence, for comprehensive clarifications, we repeat our core experiments and supplement the requested analysis on currently available L20 GPUs. All the new results are organized in the appendix of our latest submission.
>
> - **As shown in Figure 7 in the Appendix, we calculate the per-valid-token latency** by dividing the total latency by the number of accepted tokens in each sample, which is then averaged across all samples and evaluation datasets. Notably, the decode stage accounts for the majority of the time latency when compared to the prefill stage. With the rapid drafting capability and parallel verification, QSpec achieves significantly lower latency than W4A16, ranging from 28.5% to 39.7%. In detail, QSpec spends more time in the draft phase than in the high-precision verify phase. This may be attributed to the high acceptance rate of QSpec, which resulted in fewer verify requests.
>
> Hope that we can fully address your concerns, and that you can reconsider our scores! If you still have any questions, please feel free to let us know!
>
> Sincerely,
>
> Authors
>
>
>
> References
>
>
> [1] Tanishq Kumar, Zachary Ankner, Benjamin F Spector, Blake Bordelon, Niklas Muennighoff, Mansheej Paul, Cengiz Pehlevan, Christopher R´e, and Aditi Raghunathan. Scaling laws for precision. arXiv preprint arXiv:2411.04330, 2024. URL https://arxiv.org/abs/2411.04330.
>
> [2] Tianle Cai, Yuhong Li, Zhengyang Geng, Hongwu Peng, Jason D. Lee, Deming Chen, and Tri Dao. Medusa: Simple llm inference acceleration framework with multiple decoding heads, 2024. URL https://arxiv.org/abs/2401.10774.
>
> [3] Yuhui Li, Fangyun Wei, Chao Zhang, and Hongyang Zhang. Eagle: Speculative sampling requires rethinking feature uncertainty, 2024. URL https://arxiv.org/abs/2401.15077.
>
> [4] Chen, J., Tiwari, V., Sadhukhan, R., Chen, Z., Shi, J., Yen, I.E., & Chen, B. (2024). MagicDec: Breaking the Latency-Throughput Tradeoff for Long Context Generation with Speculative Decoding. ArXiv, abs/2408.11049. URL https://arxiv.org/abs/2408.11049.

---

> ### Author Response · Authors · 2024-11-23
> **Response to reviewer (4/n)**
>
> In response to your interest in the acceptance rates of different quantization methods, we further evaluate Atom and QuaRot across different datasets. The results, presented in the table below, indicate that both quantization schemes achieve very high acceptance rates, with Atom slightly outperforming QuaRot. In contrast, acceptance rates of 28–58% are observed on a 160m-7b draft-target model pair with γ=5 in conventional speculative decoding [1]. Together with the results in Figure 5 and 8 in our updated submission, QSpec consistently achieves high acceptance rates across diverse draft token length, quantization methods and datasets due to its design of shared weights and KV cache, eliminating the need of further finetuning. Given the efficiency-performance decoupling brought by QSpec, this natively high acceptance rate may illuminate the future development of more efficient quantization methods.
>
> | Quantization | lmsys | sharegpt | gsm8k | MATH  | MBPP  | HumanEval |
> |--------------|-------|----------|-------|-------|-------|-----------|
> | Atom         | 0.867 | 0.838    | 0.915 | 0.894 | 0.886 | 0.914     |
> | QuaRot       | 0.830 | 0.816    | 0.890 | 0.889 | 0.854 | 0.882     |
>
> Hope that this could further address your concerns!
>
>
> References
>
> [1] Xiaoxuan Liu, Lanxiang Hu, Peter Bailis, Alvin Cheung, Zhijie Deng, Ion Stoica, and Hao Zhang. Online speculative decoding, 2024. URL https://arxiv.org/abs/2310.07177.

---

> > ### Comment · Reviewer_PcTG · 2024-11-26
> >
> > Thanks for the detailed response. I have some further questions.
> >
> > - It is generally believed that W4A16 runs faster than W16A16 for LLMs. The results in AWQ [1] also shows  W4A16 outperforms W16A16. Could you provide more thorough analysis?
> >
> > - Could you please provide the speed results for W16A16, W4A16, W4A4 for normal llama2-7B and llama3.2-3B, without speculative decoding?
> >
> > - Why the draft model runs faster than W4A4 model? According to my understanding, the W4A4 model is used as the draft.
> >
> > Further discussion about comparison with other speculative methods
> >
> > - To regard QSpec as a new quantization paradigm seems a good point. But it still need to be discussed what's the advantages of QSpec to previous speculative decoding methods, especially considering the wall-clock speedup.
> > - Moreover, any LLM speedup methods can be used as a draft model, such as pruning, etc.
> > - The EAGLE paper also studies the combination of speculative decoding and model quantization.
> >
> > Thanks again for the detailed response.
> >
> > [1] AWQ: Activation-aware Weight Quantization for LLM Compression and Acceleration

---

> > > ### Author Response · Authors · 2024-12-02
> > > **Further response to reviewer (1/n)**
> > >
> > > Dear Reviewer PcTG,
> > >
> > > We greatly appreciate your feedbacks, and we have made thorough efforts to address your new concerns through detailed experiments.
> > >
> > > **Q1: It is generally believed that W4A16 runs faster than W16A16 for LLMs. The results in AWQ [1] also shows W4A16 outperforms W16A16. Could you provide a more thorough analysis?**
> > >
> > > We'd like to explain the conflict between our observation / experiment results w.r.t reviewer's claim by steps.
> > > 1. The observed performance differences can be attributed to the selection of baselines for comparison.
> > > - In the official implementation of AWQ, optimal performance is achieved through the use of fused layers. However, these fused layers "cannot be combined with other optimization techniques such as FlashAttention-2" [1].
> > > - ATOM lacks the AWQ fused kernel but includes state-of-the-art (SOTA) attention optimizations in its benchmark. Specifically, ATOM manages the key-value (KV) cache in blocks and utilizes FlashAttention and FlashInfer-based paged attention during runtime. As a result, (1) ATOM's AWQ implementation lacks fused layers and thus shows reduced performance, while (2) W16A16 benefits from flash/page attention optimizations, making AWQ appear relatively worse in comparison.
> > > - Conversely, the official comparison uses AWQ with fused kernels but lacks the SOTA attention optimizations. The official setup did not incorporate block-wise attention into W16A16, instead using a weaker baseline with the standard HuggingFace FP16 implementation. Since AWQ can run with fused layers, it performs relatively better in this setup. However, this approach sacrifices memory management efficiency, resulting in the official setup handling fewer requests and being (3) more prone to out-of-memory (OOM) issues.
> > > 2. W4A16 is not necessarily better than W16A16
> > > - (4) Without fused kernels, AWQ W4A16's performance can always be poorer than that of FP16 without SOTA attentions, according to the Figure https://huggingface.co/datasets/huggingface/documentation-images/resolve/main/quantization/fused_generate_throughput_plot.png [1]. Additionally, (5) even with fused kernels, AWQ is not necessarily faster than FP16 with attention optimizations, which highly depends on the input batch size. This contradicts the claim that **"W4A16 runs faster than W16A16 for LLMs."**
> > > 3. Why do we choose ATOM setup
> > > - We choose ATOM's setup since real-world serving environments (e.g., in vLLM) aligns more closely with ATOM's results, as KV cache management in blocks and attention optimizations are already common practice. Additionally, fusion for QKV in AWQ fused layer can also benefit FP16 inference, making the comparison unfair if directly comparing with the unoptimized vanilla FP16 baseline.
> > > 4. Validation Experiments
> > > - To validate our previous explanation, we benchmarked LLaMA-2-7B on an A100 GPU across batch sizes from 1 to 32, measuring token generation throughput (tokens/s). We also show that our AWQ performance aligns with ATOM's performance.
> > >
> > > Our observed results are shown the following table.
> > > | Batch Size| 1.0   | 2.0   | 4.0   | 8.0   | 16.0  | 32.0  | 64.0  |
> > > |-----------------------|--------|-------|-------|-------|-------|-------|-------|
> > > | fp16_flash           | 64.7   | 118.0 | 212.8 | 378.5 | 665.4 | 1063.0| 1532.8|
> > > | awq_flash            | 54.6   | 86.2  | 126.5 | 169.1 | 386.3 | 666.4 | 1027.5|
> > > | awq_official_origin  | 30.5   | 61.4  | 122.0 | 244.1 | 403.1 | 519.5 | oom   |
> > > | awq_fuse_layer       | 42.1   | 84.1  | 167.7 | 345.1 | 692.6 | 1233.0| oom   |
> > > | fp16_official_origin | 40.8   | 80.7  | 164.0 | 277.6 | 410.9 | 528.3 | oom   |
> > >
> > > We can see the following results (number echo to previous claim (x)):
> > > 1. Throughput: awq_official_origin < awq_fuse_layer
> > > 2. Throughput: fp_16_official_origin < fp16_flash.
> > > 3. Official implementations are OOM at BS=64 while ATOM's not.
> > > 4. Throughput: awq_official_origin < fp16_official_orgin.
> > > 5. Throughput: awq_fused_layer performs worse than fp16_flash with bs<=8
> > >
> > > In previous table, the AWQ latency reproduces the latency curve of the AWQ  vs. FP16 in the ATOM paper [2] (Figure.10), confirming its correctness.

---

> ### Author Response · Authors · 2024-12-02
> **Further response to reviewer (2/n)**
>
> **Q2:Could you please provide the speed results for W16A16, W4A16, W4A4 for normal llama2-7B and llama3.2-3B, without speculative decoding?**
> - We've already included them in the Table. 5. Check out our extended experiments for the L20. In experiment, we release the memory constraints for W16A16 via model partition and pipeline paralleism.
>
> **Q3:Why the draft model runs faster than W4A4 model? According to my understanding, the W4A4 model is used as the draft.**
> - You may refer to the Figure 7. Some of the tokens are generated by the W4A16, since each validation will generate a new token afterwards. In that way QSPEC walltime roughly composed of 3/4 W4A4 time and 1/4 W4A16 time.
>
> **Discussion 1: Comparison with EAGLE**
> - To comply with the reviewer's request, we conducted extensive experiments to compare EAGLE-Quant (using an FP16 draft model with W4A16 EAGLE), QSPEC, W4A16, and W4A4 on a single A100 GPU. We utilized the Llama-2-7b-chat-hf model and tested with batch sizes of 1, 8, and 16.
> - For EAGLE-Quant, we opted to use an FP16 draft model combined with W4A16 EAGLE. This choice was made because the official EAGLE quantization (fast-gpt) does not support batching well. Additionally, when we attempted to quantize the EAGLE draft model using GPTQ, we observed a significant deterioration in the acceptance rate. We retained the FP16 draft model for EAGLE-Quant, which is the optimal case we observed, in our extended experiments.
>
> | Quantization | Batch Size | gsm8k (8-shot) | MATH (4-shot) | MBPP (0-shot) | HumanEval (0-shot) | ShareGPT | Lmsys-1k |
> |--------------|------------|----------------|---------------|---------------|---------------------|----------|----------|
> | **Eagle**    | 1          |  50.0         |   53.8        |   51.9        |   43.0              |  60.9    |  53.3    |
> |              | 8          | 135.2         |  196.6        |  126.5        |  114.9              | 216.7    | 158.0    |
> |              | 16         | oom           |   oom         |  134.0        |   94.8              | 231.3    |   oom    |
> | **QSpec**    | 1          |  33.0         |   35.0        |   34.0        |   29.3              |  42.3    |  40.8    |
> |              | 8          | 156.5         |  196.2        |  152.4        |  124.5              | 260.9    | 193.4    |
> |              | 16         | 239.3         |  302.3        |  211.4        |  184.8              | 457.7    | 355.3    |
> | **W4A16**    | 1          |  38.4         |   42.9        |   40.4        |   34.4              |  54.3    |  52.2    |
> |              | 8          | 111.0         |  138.2        |  122.0        |  108.2              | 175.5    | 147.8    |
> |              | 16         | 170.2         |  229.5        |  190.8        |  158.1              | 400.3    | 310.4    |
> | **W4A4**     | 1          |  40.6         |   45.3        |   39.0        |   34.5              |  54.9    |  52.5    |
> |              | 8          | 200.1         |  268.1        |  173.0        |  159.5              | 363.4    | 282.3    |
> |              | 16         | 292.4         |  399.4        |  260.2        |  224.8              | 657.5    | 513.5    |
>
>
>
> - In previous table, EAGLE performs optimally with single-sequence inputs (batch size = 1) across all datasets, as noted in their paper. **However, its performance significantly deteriorates with larger batch sizes (e.g., 8 or 16)**. This is primarily because EAGLE's high acceptance rate relies heavily on a tree structure whose size scales with the batch size. The necessity of this tree structure arises from the fact that EAGLE's draft model is largely built around an additional decode block specifically trained for drafting which lacks high consistency with the target model, making the tree structure essential for bridging discrepancies and maintaining alignment. As the batch size increases, the computation load on the target model increases due to the scaled tree structure. The computation on the target model shifts from being memory-bound to compute-bound, leading to performance degradation under W4A16.
> - Furthermore, as we mentioned in our paper, speculative methods like EAGLE can **face out-of-memory (OOM) issues**—for instance, we observed that EAGLE encounters OOM at a batch size of 16. This is because the key-value (KV) storage requirements of EAGLE's draft model grow substantially with larger batches, causing OOM problems.
> - In contrast, as noted in our previous response, QSPEC is both more batch-friendly and memory-efficient. In QSPEC's drafting phase (W4A4) **does not require a tree structure to maintain a high acceptance rate**, so the workload does not scale with tree size as the batch size increases. Additionally, QSPEC does not introduce any extra memory requirements compared to the high-precision method, as we share the weights and KV cache with W4A16. As a result, QSPEC demonstrates better performance at larger batch sizes (8 and 16) compared to EAGLE, but also doesn't have the OOM issue.

---

> > ### Author Response · Authors · 2024-12-02
> > **Further response to reviewer (3/n)**
> >
> > **Discussion 2: Moreover, any LLM speedup method can be used as a draft model, such as pruning, etc.**
> > - You're correct; however, we'd like to point out that pruning also requires retraining to improve performance, effectively creating a separate draft model. Also, sparse weight/activation are often less hardware-friendly, degrades the utilization of the GPU cores. Consequently, it faces the same out-of-memory (OOM) issues we mentioned earlier and may require additional structures/kernels to maintain an acceptable acceptance rate and speed.
> >
> > **Discussion 3: The EAGLE paper also studies the combination of speculative decoding and model quantization.**
> > - We have discussed this in the previous experiment. Fast-GPT doesn't support batching, and using GPTQ significantly degrades the acceptance rate.
> >
> > References
> >
> > [1] Hugging Face. (n.d.). AWQ. Transformers Documentation. Retrieved December 1, 2024, from https://huggingface.co/docs/transformers/main/quantization/awq?fuse=supported+architectures.
> >
> > [2] Zhao, Y., Lin, C. Y., Zhu, K., Ye, Z., Chen, L., Zheng, S., Ceze, L., Krishnamurthy, A., Chen, T., & Kasikci, B. (2024). Atom: Low-bit quantization for efficient and accurate LLM serving. arXiv preprint arXiv:2310.19102. https://arxiv.org/abs/2310.19102

---

> ### Comment · Reviewer_PcTG · 2024-12-03
>
> Thank you for the detailed reply. Most of my concerns are addressed. I decide to raise my score.

---

> > ### Author Response · Authors · 2024-12-04
> > **Further response to reviewer (4/n)**
> >
> > Dear Reviewer PcTG,
> >
> > We sincerely appreciate your detailed feedback and are truly grateful for your decision to raise the score. Your constructive suggestions have been instrumental in helping us refine our work.
> >
> > Sincerely,
> >
> > Authors

---

### Meta-Review · Area_Chair_fB9i · 2024-12-20

**Metareview:**

This paper introduces QSPEC, a method that combines low-precision (W4A4) and high-precision (W4A16) quantization schemes to improve token generation throughput in large language models (LLMs) through speculative decoding. The method drafts tokens with low-precision quantization and verifies them with high-precision quantization, offering up to 1.78x throughput improvement without sacrificing output quality. The approach is claimed to be memory-efficient, leveraging shared weights and key-value caches, and does not require retraining, making it suitable for memory-constrained environments.

The paper demonstrates clear performance gains in token generation throughput, making it a relevant contribution for improving the efficiency of LLMs. The empirical results are promising, showing improvements across various tasks, model sizes, and batch sizes. Additionally, the method avoids increasing memory consumption by reusing weights and key-value caches, which is beneficial for deployment in resource-limited scenarios. The fact that the method does not require retraining is another practical advantage, facilitating easy adoption.

Reviewers raised concerns about the limited novelty of the paper, as it primarily integrates existing quantization techniques (W4A4 and W4A16) with speculative decoding. They also requested more robust comparisons, particularly with other low-bitwidth quantization methods like EAGLE and W4A4, and more detailed performance metrics (e.g., speedup vs. batch size vs. quality). Although the authors provided a detailed rebuttal and made some revisions, these concerns were not fully addressed.

The paper lacks significant novelty, relying on existing quantization techniques and speculative decoding without introducing fundamentally new methods. While the empirical results are promising, the comparisons with other methods remain insufficient, and the revisions did not fully address the key concerns raised by the reviewers. Given the highly competitive nature of the review process this year and the unresolved issues, the paper does not meet the acceptance criteria and is therefore rejected.

**Additional Comments On Reviewer Discussion:**

During the rebuttal period, the reviewers raised several key points, primarily focusing on the novelty and comparative analysis of the proposed method. One major concern was that the integration of low-precision (W4A4) and high-precision (W4A16) quantization schemes with speculative decoding did not introduce sufficiently novel ideas, as it mainly repurposed existing techniques. Additionally, reviewers requested more detailed comparisons with other low-bitwidth methods, particularly in terms of performance benchmarks such as speedup versus batch size and memory consumption. They also suggested using larger-scale benchmarks (e.g., LLM-QBench) to better demonstrate the method's effectiveness.

The authors acknowledged the concerns and made revisions, including offering additional discussions on memory efficiency and revising the experimental setup. However, the authors did not fully address the requests for clearer comparative analysis or provide the necessary charts and tables to better highlight the advantages of their method over other speculative decoding techniques. While the revisions improved the clarity of the paper, they were not sufficient to overcome the fundamental issues raised about the novelty of the approach and the lack of a robust comparative evaluation.

In my final decision, I weighed the lack of novelty and insufficient responses to the key concerns as more significant than the improvements made during the rebuttal period. Given the competitive nature of the review cycle and the fact that the critical points were not fully addressed, I recommend rejection.

---

### Decision · Program_Chairs · 2025-01-22

Reject